# Accordion-Thinking: Self-Regulated Step Summaries for Efficient and Readable LLM Reasoning

**Zhicheng Yang** [1]  **Zhijiang Guo** [1 2]  **Yinya Huang** [3]  **Yongxin Wang** [4]  **Wenlei Shi** [5]  **Yiwei Wang** [6]
**Xiaodan Liang** [4 7]  **Jing Tang** [1 2]

`yangzhch6@gmail.com`
*Project Repo*: `https://github.com/yangzhch6/Accordion-Thinking`

## Abstract

Scaling test-time compute via long Chain-of-Thought unlocks remarkable gains in reasoning capabilities, yet it faces practical limits due to the linear growth of KV cache and quadratic attention complexity. In this paper, we introduce `Accordion-Thinking`, an end-to-end framework where LLMs learn to self-regulate the granularity of the reasoning steps through dynamic summarization. This mechanism enables a *Fold* inference mode, where the model periodically summarizes its thought process and discards former thoughts to reduce dependency on historical tokens. We apply reinforcement learning to incentivize this capability further, uncovering a critical insight: the accuracy gap between the highly efficient *Fold* mode and the exhaustive *Unfold* mode progressively narrows and eventually vanishes over the course of training. This phenomenon demonstrates that the model learns to encode essential reasoning information into compact summaries, achieving effective compression of the reasoning context. Our `Accordion-Thinking` demonstrates that with learned self-compression, LLMs can tackle complex reasoning tasks with minimal dependency token overhead without compromising solution quality, and it achieves a 3× throughput while maintaining accuracy on a 48GB GPU memory configuration, while the structured step summaries provide a human-readable account of the reasoning process.

## 1. Introduction

Recent advances in Large Language Models (LLMs) have demonstrated that scaling test-time computation through long Chain-of-Thought (CoT) reasoning can dramati-

cally enhance performance on complex problem-solving tasks (Wei et al., 2022). Methods such as o1-like thinking (Jaech et al., 2024; Guo et al., 2025) exemplify this trend, where models generate extended reasoning traces that often span tens of thousands of tokens through iterative reflection, backtracking, and self-correction. However, such long-form reasoning inherently produces verbose and often unstructured internal thought processes, which not only hinder human readability but also impose significant computational burdens. Specifically, the linear growth of the KV cache and quadratic attention complexity with respect to context length limit the practical scalability of reasoning models, leading to prohibitive memory and computational costs in both training and inference.

Prior work has explored compressing intermediate thoughts, either through heuristic token eviction (Zhang et al., 2025) or fixed-length chunking with state carryover (Aghajohari et al., 2025). However, such approaches often rely on external heuristics or rule-based segmentation, which may disrupt the natural flow of reasoning and fail to improve the readability of reasoning traces. Moreover, they typically treat compression as a separate stage or a static hyperparameter, rather than as a learnable capability integrated into the model's reasoning process.

In this paper, we revisit the problem from a self-regulatory perspective: rather than imposing compression schedules externally, we enable the LLM to learn *when* and *how* to summarize its own reasoning process dynamically. We introduce `Accordion-Thinking`, an end-to-end training framework in which the model learns to alternate between detailed reasoning steps and compact summaries, thereby reducing its reliance on long token histories without compromising reasoning integrity. Inspired by the human ability to condense complex thoughts into concise summaries while retaining logical continuity, our approach is built on a simple but powerful insight: reasoning and summarization are complementary skills that can be jointly cultivated through reinforcement learning.

We therefore conduct a systematic study of post-training strategies for `Accordion-Thinking`, starting from a base language model without prior compression-specific tuning. To instill foldable capability, we synthetically aug-

---

[1]HKUST(GZ) [2]HKUST [3]ETH AI Center, ETH Zurich [4]MBZUAI [5]ByteDance, Seed [6]University of California, Merced [7]Sun Yat-Sen University. Correspondence to: Jing Tang <jing-tang@ust.hk>.

*Proceedings of the 43rd International Conference on Machine Learning*, Seoul, South Korea. PMLR 306, 2026. Copyright 2026 by the author(s).

ment standard CoT data into a structured format where each reasoning segment is followed by a concise summary formatted in `<step>...</step>`, training the model to produce and later rely on these self-generated summaries during extended inference. However, supervised fine-tuning alone is insufficient for robust generalization. We therefore employ Reinforcement Learning to further incentivize efficient and accurate compression behavior. We compare three training regimes: (1) standard long-context reasoning (*Unfold* mode), (2) compressed-step reasoning with periodic summarization (*Fold* mode), and (3) a mixed regime that interleaves both. Crucially, we observe a "Gap-Vanishing" phenomenon: while the *Fold* mode initially lags behind the full-context *Unfold* mode, the performance gap between the two gradually disappears as RL training progresses. This convergence indicates that the model successfully learns to preserve essential reasoning information even in compressed form, making the *Fold* mode a viable, high-efficiency alternative to standard CoT. Moreover, the structured step summaries produced by Accordion-Thinking provide a clear and readable account of the model's reasoning traces. We observe that the sequence of summaries alone can serve as a faithful substitute for the final solution, providing users with immediate insight into how the answer was derived. We consider this is a significant step towards efficient and transparent reasoning systems.

Our contributions are summarized as follows:

- We propose the **`Accordion-Thinking` framework**, which includes a data synthesis pipeline and an RL training pipeline that instills self-regulated compression to teach LLMs to generate and use step-wise summaries.

- We systematically discuss and eliminate the **Performance Degradation** caused by foldable CoT reasoning. We thereby explore post-training strategies for `Accordion-Thinking` and reveal that the Performance Gap between the compressed *Fold* mode and the full-context *Unfold* mode vanishes over training, validating the model's ability to perform effective self-compression. The experimental results show that our method not only eliminates the gap but even surpasses the performance of vanilla CoT.

- **Efficiency and Readability**. Experimental analyses across multiple reasoning benchmarks show that our method achieves triple the throughput under limited GPU memory conditions, reducing dependency on historical tokens, validating its efficiency and for complex reasoning tasks. Human evaluation confirms that the generated step summaries are coherent and semantically faithful, often serving as direct substitutes for the final answer.

## 2. Related Works

### 2.1. Slow Thinking

Complex reasoning tasks (He et al., 2024; Lewkowycz et al., 2022; Zeng et al., 2024; Yang et al., 2025c; Xiang et al., 2025), such as mathematical problem solving, are one of the most challenging tasks for LLMs, necessitating a shift from fast-thinking to slow-thinking. Chain-of-Thoughts (Wei et al., 2022) teaches the LLM to decompose complex questions and solve them step-by-step. Based on this, OpenAI-o1 (Jaech et al., 2024), DeepSeek-R1 (Guo et al., 2025), and Kimi-1.5 (Team et al., 2025), has pushed the frontier of LLM capability, especially for demanding tasks in complex reasoning such as mathematics and programming. Following early approaches that trained reward models from preference data (Ouyang et al., 2022), subsequent methods like Direct Preference Optimization (Rafailov et al., 2023) offered a more streamlined alternative by optimizing policy objectives directly on pairwise comparisons. Recent advancements have shifted focus towards training with Reinforcement Learning under Verifiable Rewards (RLVR), a paradigm exemplified by DeepSeek-R1 (Guo et al., 2025) which aims to cultivate a model's capacity for deliberate, multi-step reasoning. However, emerging analysis (Liu et al., 2025a; Zhao et al., 2025a; Shah et al., 2025) suggests that the self-improvement behaviors elicited by such training may be inherent capabilities of the base model, unlocked rather than created by the RL process. Algorithmically, Group Relative Policy Optimization (GRPO) (Shao et al., 2024) has become a prominent RLVR technique. DARS (Yang et al., 2025b) is proposed to work as a focal loss in GPRO. Building on PPO (Schulman et al., 2017), group-relative advantage estimation has inspired several variants, including DAPO (Yu et al., 2025), VAPO (Yue et al., 2025), and Dr. GRPO (Liu et al., 2025b).

### 2.2. Efficient Reasoning

Prior research has pursued efficiency through various strategies. Several approaches distill reasoning processes by omitting intermediate steps or tokens (Liu et al., 2024a; Xia et al., 2025; Li et al., 2025). Alternative methods dynamically manage length via early-exit mechanisms (Ding et al., 2024), certainty guided generation (Huang et al., 2025), allocate token budgets adaptively based on problem difficulty (Han et al., 2025), or guide model activations toward more concise outputs (Zhao et al., 2025b). Structured prompting and collaborative frameworks also contribute to token reduction; for instance, CoThinking first outlines a plan before reasoning (Fan et al., 2025). Similarly, Cheng & Van Durme (2024) employs shorter traces of contemplative tokens to streamline reasoning. Another line of work approximates attention computations during inference by modifying attention mechanisms and masking less critical tokens. For example, Zhang et al. (2023); Yang et al. (2024b) identify

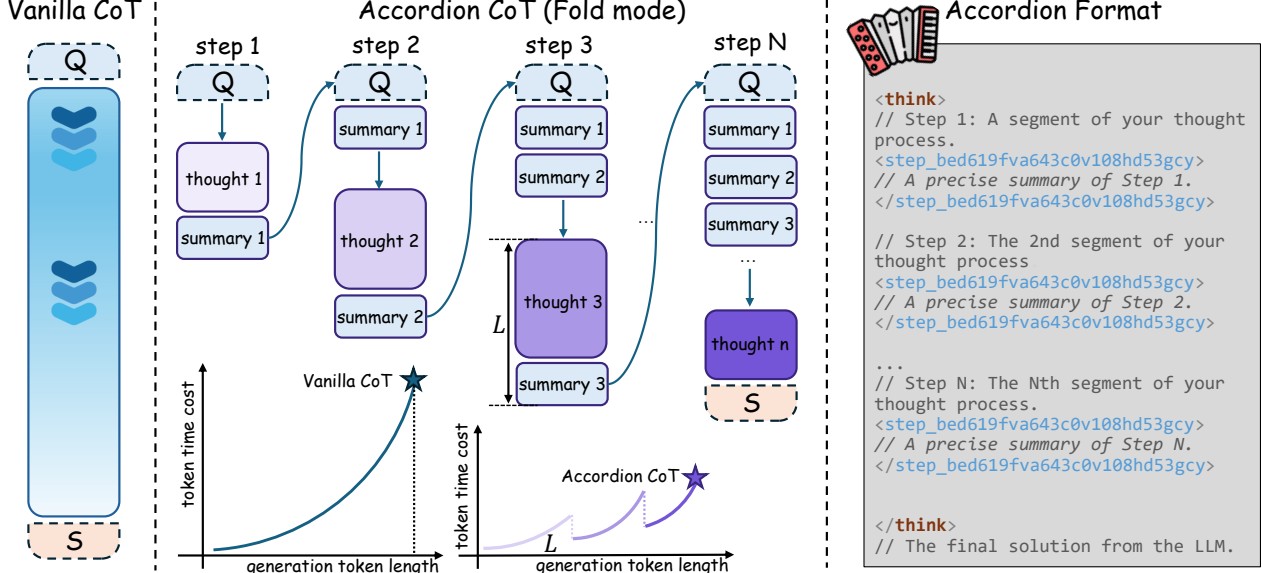

*Figure 1.* Comparison of Vanilla CoT and our Accordion CoT. As the generation length increases, the computational complexity per token in Vanilla CoT grows quadratically. In contrast, our Accordion CoT folds the context after each step, reducing the computational complexity for the next token generation and improving inference speed. We force the model to follow the Accordion Format, which splits the whole thinking process into several coarse level steps followed by a readable summary. We add 2 special tokens to the model vocabulary. Each generation stops at `</step>` or the EOS token.

and retain tokens with high estimated attention contributions, whereas Xiao et al. (2023) maintains a limited set of attention-sink tokens to preserve stability under sliding-window contexts. Complementary compression techniques reduce the memory footprint of each retained token via aggressive quantization without sacrificing accuracy (Hooper et al., 2024; Liu et al., 2024b). More recent efforts leverage existing model parameters to learn token-eviction policies during inference, typically through distillation on original model predictions combined with sparsity regularization (Łańcucki et al., 2025). LighterThinker (Zhang et al., 2025) compresses verbose thought steps into compact representations and discards the original reasoning chains. InftyThink (Yan et al., 2025) synthesizes a huge amount of summarize-and-continue data to conduct heavy SFT, without delving into RL methods and the performance loss caused by summarize SFT. DeleThink (Aghajohari et al., 2025) structures the reasoning process into fixed-size chunks, and abandons the first half chunk in each generation step. Unlike previous approaches, we propose `Accordion-Thinking`, which allows the model to perform progressive step-by-step reasoning by retaining the summary of its current reasoning at each step before proceeding further. We further reveal the performance degradation caused by summary and eliminate the gap with `Accordion-Thinking` framework.

## 3. Method: `Accordion-Thinking`

### 3.1. Problem Formulation

We formulate the reasoning process as a sequential generation task. Let $\mathbf{x}$ denote the input query and $\mathbf{a}$ denote the final answer. In standard Chain-of-Thought (CoT) reasoning, the model generates a reasoning chain $\mathbf{r}$ before predicting $\mathbf{a}$. We propose to structure this chain into $K$ discrete steps.

**Accordion Structure.** Each reasoning step $k \in \{1, \ldots, K\}$ consists of a tuple $(\mathbf{d}_k, \mathbf{s}_k)$, where:

- $\mathbf{d}_k$: The **detailed reasoning** segment, containing the free-form exploration and derivation.

- $\mathbf{s}_k$: The **step summary**, a concise abstraction of the state updates and logical conclusions derived in $\mathbf{d}_k$.

The full sequence is thus $\mathbf{y} = [\mathbf{x}, \mathbf{d}_1, \mathbf{s}_1, \ldots, \mathbf{d}_K, \mathbf{s}_K, \mathbf{a}]$. Special control tokens (e.g., `<step>`) are used to delineate these segments in implementation, but omitted here for notational clarity.

**Context Definitions.** The core distinction between standard CoT and our approach lies in the *visible history* (context) available to the model when generating the next segment. Let $\mathcal{H}_k$ denote the context used to generate the $k$-th detailed segment $\mathbf{d}_k$.

**1. Unfold Mode (Full-Context).** This corresponds to standard CoT, where the model attends to the complete history of all previous details and summaries. The context at step $k$ is:

$$\mathcal{H}_k^{\text{unfold}} = [\mathbf{x}, \mathbf{d}_1, \mathbf{s}_1, \ldots, \mathbf{d}_{k-1}, \mathbf{s}_{k-1}] \quad (1)$$

The computational complexity for attention at step $K$ scales with $O(|\mathbf{x}| + \sum_{i=1}^{K-1}(|\mathbf{d}_i| + |\mathbf{s}_i|))$.

**2. Fold Mode (Compressed-Context).** In this mode, we enforce a dynamic context pruning mechanism. Once a summary $\mathbf{s}_{k-1}$ is generated, the corresponding detailed derivation $\mathbf{d}_{k-1}$ is discarded from the KV cache. The context retains only the input and the sequence of past summaries:

$$\mathcal{H}_k^{\text{fold}} = [\mathbf{x}, \mathbf{s}_1, \mathbf{s}_2, \ldots, \mathbf{s}_{k-1}] \tag{2}$$

Consequently, the generation of the next reasoning step is modeled as:

$$P(\mathbf{d}_k \mid \cdot) = \pi_\theta(\mathbf{d}_k \mid \mathcal{H}_k^{\text{fold}}) \tag{3}$$

Crucially, when generating the summary $\mathbf{s}_k$ immediately following $\mathbf{d}_k$, the detailed segment $\mathbf{d}_k$ remains temporarily visible to ensure the summary faithfully captures the just-derived logic. The folding operation (pruning $\mathbf{d}_k$) occurs strictly after $\mathbf{s}_k$ is completed.

This formulation reduces the memory complexity to $O(|\mathbf{x}| + \sum_{i=1}^{K-1} |\mathbf{s}_i|)$. Since $|\mathbf{s}_i| \ll |\mathbf{d}_i|$, *Fold* mode significantly extends the effective context window and inference throughput.

### 3.2. Accordion Data Synthesis

To instill the ability of self-summarization and stepwise reasoning, we construct a synthetic training dataset with a segmented format and explicit step summaries, this process is similar to InftyThink (Yan et al., 2025). The goal is to teach the model to produce and later rely on concise, structured summaries of its reasoning process, enabling both efficient inference (Fold mode) and human-readable reasoning traces. However, we further show that SFT alone is insufficient; reinforcement learning is necessary to enable LLMs to truly master reasoning in the folding pattern. The pipeline consists of three stages

1. **Seed Data Collection.** We randomly sample 10,000 reasoning traces from the `openr1-math-46k` dataset (Qu et al., 2025), a large-scale collection of long-form CoT examples within 16k response length. Each seed example contains a query $X$ and a long reasoning trace $R_{\text{raw}}$ followed by a final answer $A$.

2. **Structured Rewriting via Teacher LLM.** For each seed pair $(X, R_{\text{raw}})$, we prompt DeepSeek-V3.2 (DeepSeek-AI et al., 2025) to rewrite the free-form reasoning trace into our structured Accordion format as shown in Figure 1. The prompt template is shown in I.

3. **Rule-based Filter.** To ensure high-quality training data, we apply the following criteria to each rewritten example. Examples that fail any of the following criteria are discarded:
   - **Structural integrity**: Each step must be properly enclosed by `<step>` and `</step>` tags, and the entire trace must be wrapped in `<think>...</think>`.

   - **Step count and length**: The total number of steps $K$ must be between 2 and 6 (inclusive) to avoid overly fragmented or monolithic reasoning. Each detailed reasoning block must not exceed 6,144 tokens to prevent excessively verbose steps.
   - **Summary length**: Summary must contain at least 100 tokens, encouraging sufficiently informative compression. We hypothesize that richer summaries provide better support for subsequent reasoning in Fold mode.

We collect 3,900 samples with the above pipeline, and then convert them to *Fold* mode with 14,653 samples. We provide additional ablation studies on the data synthesis pipeline in Section 4.3. The synthetic dataset is combined with the original `openr1-math-46k` to cold-start the base models.

### 3.3. Accordion Reinforcement Learning

Supervised Fine-Tuning (SFT) effectively aligns the model with the structural requirements of Accordion-Thinking (i.e., generating `<step>` tags). However, SFT alone is insufficient to guarantee that the generated summaries are semantically complete. In the *Fold* mode, the model must learn to compress all necessary historical state information into the summary, as the detailed reasoning trace is discarded. If the summary is lossy, subsequent reasoning steps will fail.

To address this, we employ Reinforcement Learning to incentivize the model to generate high-quality, self-contained summaries that support robust reasoning under compressed contexts. We posit that the ability to reason (*generating the solution*) and the ability to compress (*summarizing the state*) are mutually reinforcing skills.

#### 3.3.1. OPTIMIZATION OBJECTIVE

We adopt the clipped objective of GRPO without the KL penalty term. Following Dr. GRPO, we likewise remove the response length handling from the GRPO target. Specifically, for a problem $q$ sampled in training data $\mathcal{D}$, the training target is formalized as:

$$\mathcal{J}(\theta) = \mathbb{E}_{(q \sim \mathcal{D}, \{o_i\}_{i=1}^{G} \sim \pi_{\theta_{\text{old}}}(q))} \left[ \frac{1}{G} \sum_{i=1}^{G} \sum_{t=1}^{|o_i|} \left( \right. \right.$$
$$\left. \left. \min\left( r_{i,t}(\theta)\hat{A}_{i,t}, \text{clip}\left(r_{i,t}(\theta), 1-\varepsilon, 1+\varepsilon\right)\hat{A}_{i,t} \right) \right) \right], \tag{4}$$

where

$$r_{i,t}(\theta) = \frac{\pi_\theta(o_{i,t} \mid q, o_{i,<t})}{\pi_{\theta_{\text{old}}}(o_{i,t} \mid q, o_{i,<t})}. \tag{5}$$

The token advantage $\hat{A}_{i,t}$ is computed using Equation 6.

$$\hat{A}_i = r_i - u, \tag{6}$$

Here, $r_i \in {0, 1}$ is the binary, trajectory-level verifiable reward for the $i$-th generated output $o_i$ (e.g., final answer

correctness). $u$ is the mean rewards across the group of $G$ samples generated for the same query $q$.

### 3.3.2. DYNAMIC CONTEXT PRUNING AND TRAINING STRATEGIES

Unlike standard RLHF, which operates on static full sequences, Accordion-RL introduces a dynamic environment where the context window changes based on the model's own outputs. We define three training strategies:

**1. Unfold Mode (Full Context Baseline).** In this setting, the model generates the full sequence with access to the entire history. The context $\mathcal{C}_{i,<t}$ simply includes the query and all previous tokens $[q, o_{i,<t}]$. In practice, the rule filter in Section 3.2 performs a format check on the rollout.

**2. Fold Mode (Compressed Context).** To enforce efficient state tracking, we implement the *Fold* mode during rollout generation. The generation process is constrained by a maximum number of steps $N$ and a maximum token length per step $L$. Let the output stream be divided into segments $S_1, S_2, \ldots$ . When the model generates the closing tag `<step>`, the environment triggers a *Fold* operation:

1. The detailed reasoning content within the current step is identified.

2. The context is updated to retain only the query and the sequence of summaries generated so far.

Consequently, when generating the next step, the policy $\pi_\theta$ conditions only on the compressed history. This forces the model to encode all critical logic into the summary block. If the summary is insufficient, the model loses the context required to solve the problem, leading to a zero reward. The reward $r_i$ is set to 1 only when the rollout $o_i$ correctly solves the problem and passes the rule filter; otherwise, it is set to 0. Furthermore, all steps within a rollout share the same reward. This hard constraint serves as a strong signal for learning effective summarization.

**3. Mixed-Mode Training.** To better observe the relationship between the *Fold* and *Unfold* modes, we introduced mixed training. Both modes were executed in a single training step, and then updated sequentially. Notably, we observe a "gap-vanishing" phenomenon: over the course of Mixed-Mode RL training, the accuracy gap between the highly efficient *Fold* inference and the exhaustive *Unfold* inference narrows and eventually disappears, indicating that the model has successfully internalized the ability to compress reasoning without information loss. It also indicates that both modes can be optimized simultaneously.

## 4. Experiments

### 4.1. Setup

**Evaluation and Training Data:** We evaluate our models using 5 widely used mathematical reasoning benchmarks: MATH-500 (Lightman et al., 2023), OlympiadBench (He et al., 2024), MinvervaMath (Lewkowycz et al., 2022), AIME24, and AMC23. We report the *Pass@1 (Avg@32)* performance on all of the evaluation benchmarks. The training data used in this work is OpenR1-45K, which is a subset of OpenR1-Math-220k (Hugging Face, 2025).

**Implementation Details:** Our experiments are conducted with Qwen2.5-Math-7B (Yang et al., 2024a) and Qwen3-4B-Base (Yang et al., 2025a). For Qwen2.5-Math, we change the rope theta to 40000 and extend the window size to 32768. In addition, to facilitate step format detection, we added two step special tokens to the model's vocabulary, as shown in Figure 1. The model terminates generation upon encountering either the `</step>` or EOS token. For cold start SFT, the warmup ratio is 0.1, the learning rate is 1e-5, and the batch size is set as 8. We train each model for 3 epochs. During the RL training, the learning rate is 1e-6, the rollout batch size is 128, and each prompt has 8 rollout generations. In practice, we do not use the reference model and KL loss. We use Math-Verify [1] as our reward function. We use temperature=1.0 for both rollout generation and evaluation. For training with *Fold* mode, the maximum number of steps $N$ is set as 6 the maximum token length per step $L$ is set as 6144. This configuration limits the model's maximum output length to 36k tokens. In practice, due to our format check mechanism, the model can hardly reach this limit.

**Baseline and Methods:** We compared the following methods: (1) *Zero-RL*: Directly using GRPO to perform zero-RL on the base model. (2) *Cold-Start*: We directly used the cold-start dataset constructed in Section 3.2 to fine-tune the base model (3) *UnFold-RL*: Conducting Unfold mode RL experiments based on the cold-start model. (4) *H2O*, *LightThinker*, and *DeleThink*: Conducting their original method on the cold-start model. (5) *Fold-RL*: Conducting Fold mode RL experiments based on the cold-start model. (6) *Mix-RL*: Conducting Mix mode RL experiments based on the cold-start model.

We denote the acquired model trained with *Mix-RL* start from Qwen2.5-Math-7B / Qwen3-4B-Base as **Accordion-Thinker**-7B / **Accordion-Thinker**-4B. In *Fold* mode, they demonstrate the same performance as *UnFold-RL* trained models, while possessing efficient CoT folding capabilities and can provide instant, readable step summaries.

---

[1] https://github.com/huggingface/Math-Verify

*Table 1.* Overall performance comparison of *Pass@1* (*Avg@32*) for Qwen2.5-Math-7B and Qwen3-4B-Base on selected benchmarks.

| Method | Gen Mode | AIME24 | AIME25 | MATH500 | AMC | Minerva | *Macro* |
|---|---|---|---|---|---|---|---|
| | | *Qwen2.5-Math-7B* | | | | | |
| Zero-RL | *Unfold* | 25.8 | 18.1 | 82.2 | 58.9 | 37.8 | 44.6 |
| | *Unfold* | 26.7 | 24.6 | 86.2 | 65.4 | 39.7 | 48.5 |
| Cold-Start | *DeleThink* | 22.5 (↓ **4.2**) | 21.1 (↓ **3.5**) | 83.7 (↓ **2.5**) | 60.6 (↓ **4.8**) | 38.4 (↓ **1.3**) | 45.3 (↓ **3.2**) |
| | *Fold* | 23.0 (↓ **3.7**) | 23.1 (↓ **1.5**) | 82.3 (↓ **3.9**) | 62.4 (↓ **3.0**) | 37.6 (↓ **2.1**) | 45.7 (↓ **2.8**) |
| *Unfold*-RL | *Unfold* | 32.0 | 26.7 | 89.2 | 71.2 | **42.1** | 52.2 |
| | *Fold* | 29.1 (↓ **2.9**) | 25.1 (↓ **1.6**) | 87.3 (↓ **1.9**) | 70.2 (↓ **1.0**) | 39.7 (↓ **2.4**) | 50.3 (↓ **1.9**) |
| H2O | *Fold* | 25.1 (↓ **6.9**) | 20.3 (↓ **6.4**) | 82.9 (↓ **7.3**) | 61.5 (↓ **9.7**) | 34.7 (↓ **7.4**) | 44.9 (↓ **7.5**) |
| LightThinker | *Fold* | 27.2 (↓ **4.8**) | 22.5 (↓ **4.2**) | 84.4 (↓ **5.8**) | 67.7 (↓ **3.5**) | 37.0 (↓ **5.1**) | 47.8 (↓ **4.6**) |
| *DeleThink* | *DeleThink* | 31.0 (↓ **1.0**) | 26.9 (↑ **0.2**) | 89.3 (↑ **0.1**) | 72.5 (↑ **1.3**) | 42.0 (↓ 0.1) | 52.3 (↑ **0.1**) |
| *Fold*-RL (**ours**) | *Fold* | 31.3 (↓ 0.7) | 26.9 (↑ **0.2**) | **89.9** (↑ **0.7**) | **73.8** (↑ **2.6**) | 42.0 (↓ 0.1) | 52.7 (↑ **0.5**) |
| *Mix*-RL (**ours**) | *Fold* | **32.2** (↑ 0.2) | **28.3** (↑ **1.6**) | 89.6 (↑ 0.4) | 71.9 (↑ 0.7) | 41.8 (↓ 0.3) | **52.8** (↑ 0.6) |
| | | *Qwen3-4B-Base* | | | | | |
| Zero-RL | *Unfold* | 25.5 | 22.5 | 85.5 | 65.4 | 39.2 | 47.6 |
| | *Unfold* | 23.8 | 25.4 | 84.7 | 64.1 | 39.5 | 47.5 |
| Cold-Start | *DeleThink* | 18.8 (↓ **5.0**) | 15.4 (↓ **10.0**) | 80.9 (↓ **3.8**) | 55.0 (↓ **9.1**) | 37.9 (↓ **1.6**) | 41.6 (↓ **5.9**) |
| | *Fold* | 19.2 (↓ **4.6**) | 22.0 (↓ **3.4**) | 79.2 (↓ **5.5**) | 57.3 (↓ **6.8**) | 35.5 (↓ **4.0**) | 42.6 (↓ **4.9**) |
| *Unfold*-RL | *Unfold* | 27.5 | 27.8 | 88.9 | **73.2** | 42.5 | 52.0 |
| | *Fold* | 25.8 (↓ **1.7**) | 25.0 (↓ **2.8**) | 85.6 (↓ **3.3**) | 69.7 (↓ **3.5**) | 39.9 (↓ **2.6**) | 49.2 (↓ **2.8**) |
| H2O | *Fold* | 22.7 (↓ **4.8**) | 20.9 (↓ **6.9**) | 80.4 (↓ **8.5**) | 64.7 (↓ **8.5**) | 34.6 (↓ **7.9**) | 44.7 (↓ **7.3**) |
| LightThinker | *Fold* | 23.9 (↓ **3.6**) | 23.2 (↓ **4.6**) | 84.0 (↓ **4.9**) | 67.7 (↓ **5.5**) | 39.0 (↓ **3.5**) | 47.6 (↓ **4.4**) |
| *DeleThink* | *DeleThink* | 25.5 (↓ **2.0**) | 26.7 (↓ **1.1**) | **89.2** (↑ **0.3**) | 72.7 (↓ 0.5) | **43.6** (↑ **1.1**) | 51.5 (↓ 0.5) |
| *Fold*-RL (**ours**) | *Fold* | **28.4** (↑ **0.9**) | 27.8 (↑ **0.0**) | 89.1 (↑ 0.2) | 72.2 (↓ 1.0) | 42.9 (↑ 0.4) | **52.1** (↑ **0.1**) |
| *Mix*-RL (**ours**) | *Fold* | 27.6 (↑ **0.1**) | **28.0** (↑ **0.2**) | 88.6 (↓ 0.3) | 72.8 (↓ 0.4) | 43.4 (↑ **0.9**) | **52.1** (↑ **0.1**) |

*Table 2.* Comparison of token efficiency under memory limit scenarios on AIME24/25.

| Model | Mode | Mem | Peak Token | Throughput |
|---|---|---|---|---|
| *Fold*-RL | *Fold* | 24Gb | 5.9k | **2971** token/s |
| *Mix*-RL | *Fold* | 24Gb | 5.7k | **3182** token/s |
| *Unfold*-RL | *Unfold* | 24Gb | 12.3k | 1083 token/s |
| *Fold*-RL | *Fold* | 48Gb | 5.9k | **5612** token/s |
| *Mix*-RL | *Fold* | 48Gb | 5.7k | **5888** token/s |
| *Unfold*-RL | Unfold | 48Gb | 12.3k | 1483 token/s |

## 4.2. Main Results

Table 1 presents the comprehensive evaluation results across two model architectures (Qwen2.5-Math-7B and Qwen3-4B-Base) on five challenging mathematical reasoning benchmarks. We report *Pass@1* with 32 samples (*Avg@32*). The results highlight three critical observations regarding the efficacy of `Accordion-Thinking`. We also investigate the inference efficiency of our **Accordion-Thinker** in this section.

**1. SFT is insufficient for robust self-compression.** We observe that the *Cold-Start* model, trained solely via Supervised Fine-Tuning on synthetic Accordion data, suffers significant performance degradation when switching from *Unfold* to *Fold* mode. On Qwen2.5-Math-7B, the average accuracy drops from 48.5% (*Unfold*) to 45.7% (*Fold*), a decline of 2.8 points. The gap is even more pronounced for Qwen3-4B-Base, with a 4.9 point drop. This indicates that while SFT teaches the model the structural format of summarization, it fails to incentivize the model to encode critical reasoning states into the summaries, leading to information loss when the detailed context is discarded.

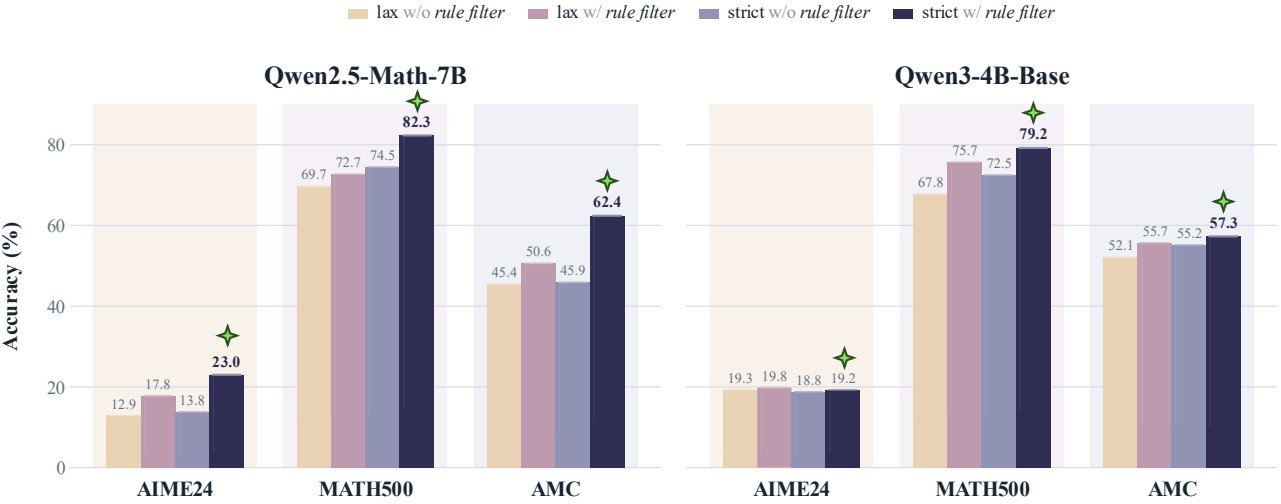

*Figure 2.* Ablation study on synthetic Accordion data for Qwen2.5-Math-7B and Qwen3-4B-Base on *Fold* mode.

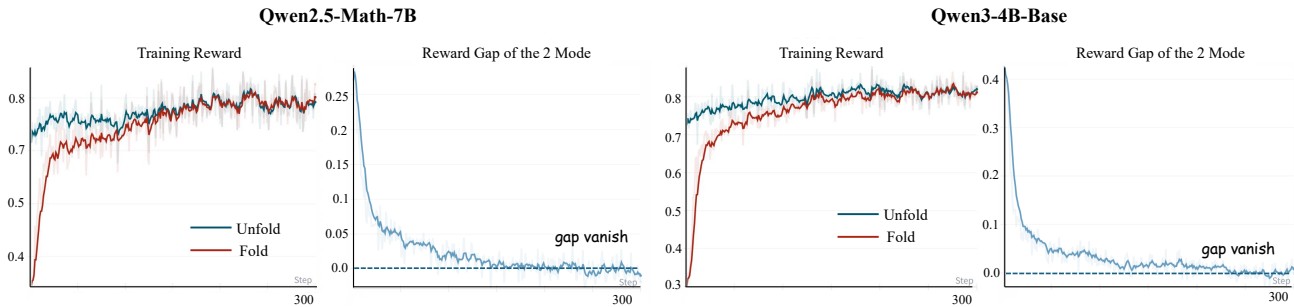

*Figure 3.* Reward gap between *Fold* mode and *Unfold* mode vanishes during *Mix*-RL training.

**2. Standard RL improves reasoning but still neglects compression.** The *Unfold*-RL baseline, which optimizes reasoning using the full context history, significantly boosts general performance compared to the Cold-Start model (e.g., 52.2% vs. 48.5% on Qwen2.5-Math-7B). However, it does not close the compression gap. When Fold-*RL* models are forced to operate in *Fold* mode during inference, they still exhibit a notable performance drop (↓1.9% on 7B and ↓2.8% on 4B). This suggests that without explicit penalties for information loss during training, the model continues to rely on the full context rather than high-quality step summaries.

**3. Accordion-Thinking achieves lossless compression.** Our proposed methods, *Fold*-RL and *Mix*-RL, successfully bridge the performance gap. Strikingly, *Fold*-RL operating in compressed mode not only recovers the performance lost in the Cold-Start phase but also matches the performance of the full-context *Unfold*-RL baseline. This demonstrates that Accordion-Thinking learns to treat summarization as an integral part of the reasoning process, enabling high-efficiency inference without compromising solution accuracy.

### 4.3. Ablation Study on Data Synthesis Pipeline

To validate the design of our data synthesis pipeline, we conduct an ablation study on two key components: the **Prompt Strategy** used for rewriting CoT traces and the **Rule-Based Filter**. We compare our proposed *Strict Prompt* as shown in Appendix I, which enforces semantic completeness and coherent segmentation, against a *Lax Prompt* that requests general segmentation without rigorous content constraints. Additionally, we evaluate the impact of applying the rule Filter (described in Section 3.2).

Figure 2 reports the *Fold* accuracy of models trained under these four configurations. The results reveal two consistent trends. First, Strict Prompting significantly outperforms Lax Prompting. Models trained with Lax prompts often generate vague summaries (e.g., "I calculated the result") that fail to preserve the reasoning state, leading to inference collapse. In contrast, the Strict prompt ensures that summaries contain sufficient information density to substitute for detailed reasoning. Second, applying the Rule-Based Filter yields consistent gains. Consequently, the combination of Strict Prompting and Filtering achieves the highest performance, confirming the necessity of high-quality supervision.

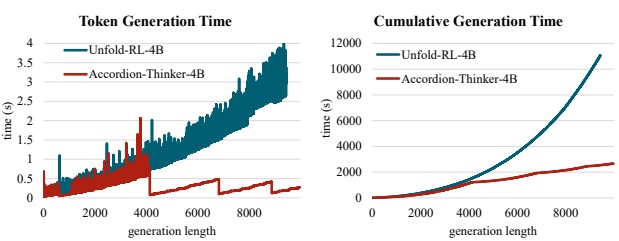

Figure 4. Comparison of token efficiency in raw PyTorch.

## 4.4. Performance Gap Vanish

To understand the dynamic relationship between full-context and compressed reasoning, we visualize the reward trajectories of both *Fold* and *Unfold* modes during the *Mix*-RL training process in Figure 3. At the onset of RL training, a significant performance gap exists between the two inference modes, with *Fold* lagging behind *Unfold* by approximately 42 points for Qwen3-4B-Base and 27 points for Qwen2.5-Math-7B. This initial disparity confirms that although the SFT stage teaches the model the structural format of Accordion-Thinking, the model has not yet learned to compress critical state information effectively, resulting in severe information loss when the reasoning trace is hidden.

As the training progresses, however, we observe a striking "Gap-Vanishing" phenomenon where the performance of the *Fold* mode improves at a significantly faster rate than that of the *Unfold* mode. The model quickly adapts to the penalty of information loss by generating higher-fidelity summaries that preserve essential logic, eventually causing the reward curves to converge and eliminating the initial gap. This synchronization suggests that the model successfully internalizes the compression mechanism, reaching a state where the sequence of step summaries carries virtually the same informational value as the full reasoning chain, thus empirically validating that Accordion-Thinking achieves effective compression through reinforcement learning.

## 4.5. Accordion-Thinking Efficiency

**Peak Token.** Peak token refers to the maximum total sequence length, calculated as the sum of prompt and response token lengths. This key metric directly governs the maximum size of the KV cache required, serving as a fundamental constraint on the model's GPU memory consumption and runtime performance. We show the training dynamics of peak token length for *Fold* and *Unfold* mode in Figure 6, **Accordion-Thinker** reduces KV Cache usage by 40% by discarding non-essential information in the chain-of-thought, without incurring a significantly larger total token count (5500 vs. 5000), thus delivering substantial efficiency improvements.

Table 3 further summarizes the benchmark-level peak token statistics on all five in-domain math benchmarks. Compared

*Table 3.* Peak token statistics (in thousands) on the five in-domain math benchmarks. (Lower is better)

| Method | AIME24 | AIME25 | MATH500 | AMC | Minerva |
|--------|--------|--------|---------|-----|---------|
| *Qwen2.5-Math-7B* | | | | | |
| *Unfold*-RL | 10.3 | 11.3 | 4.9 | 7.2 | 6.9 |
| *Fold*-RL | **4.4** | 4.5 | 2.4 | 3.1 | **2.9** |
| *Mix*-RL | **4.4** | **4.4** | **2.1** | **3.0** | **2.9** |
| *Qwen3-4B-Base* | | | | | |
| *Unfold*-RL | 12.3 | 12.9 | 6.9 | 10.3 | 9.4 |
| *Fold*-RL | 5.9 | 5.5 | 2.8 | **4.0** | 3.7 |
| *Mix*-RL | **5.7** | **5.4** | **2.7** | 4.1 | **3.6** |

with *Unfold*-RL, both *Fold*-RL and *Mix*-RL consistently reduce the peak sequence length by roughly 50%–60% across all tasks, confirming that the KV-cache savings are not limited to a single dataset.

**System Throughput.** We evaluate deployment efficiency using the vLLM engine on a single GPU, simulating memory-constrained environments (24GB/48GB) typical of high-concurrency scenarios. As shown in Table 2, Accordion-Thinking delivers substantial throughput gains, achieving a near $3\times$ speedup (5888 vs. 1483 tokens/s) for the 4B model under a 48GB limit. While standard CoT suffers from linear KV cache growth that forces reduced batch sizes or memory swapping, our *Fold* mechanism periodically discards the intermediate activation states of completed reasoning steps. This keeps the active context compact, maximizing GPU utilization and maintaining high generation speeds even during extensive reasoning chains.

**Algorithmic Scalability.** To analyze latency independent of system optimizations, we measure per-token generation time using raw PyTorch. As illustrated in Figure 4, vanilla CoT shows the $O(L^2)$ total complexity as the sequence lengthens. In contrast, **Accordion-Thinker** displays a "sawtooth" pattern where computational costs drop immediately after folding operations.

## 4.6. Accordion-Thinking Leads to Dense Attention

We further compare the attention heat maps of folded and unfolded chains of thought to examine how the model reuses intermediate reasoning during generation. Darker colors indicate that later-generated tokens rely more heavily on earlier tokens, while the exact construction of the heat map is deferred to Appendix D. As shown in Figure 5, folded reasoning consistently yields darker and more concentrated heat patterns than the unfolded baseline. This suggests that, instead of spreading attention over a long and diffuse reasoning trace, the model can rely on a smaller set of more informative folded tokens. Overall, the results indicate that folded reasoning provides a denser and more reusable representation for subsequent generations.

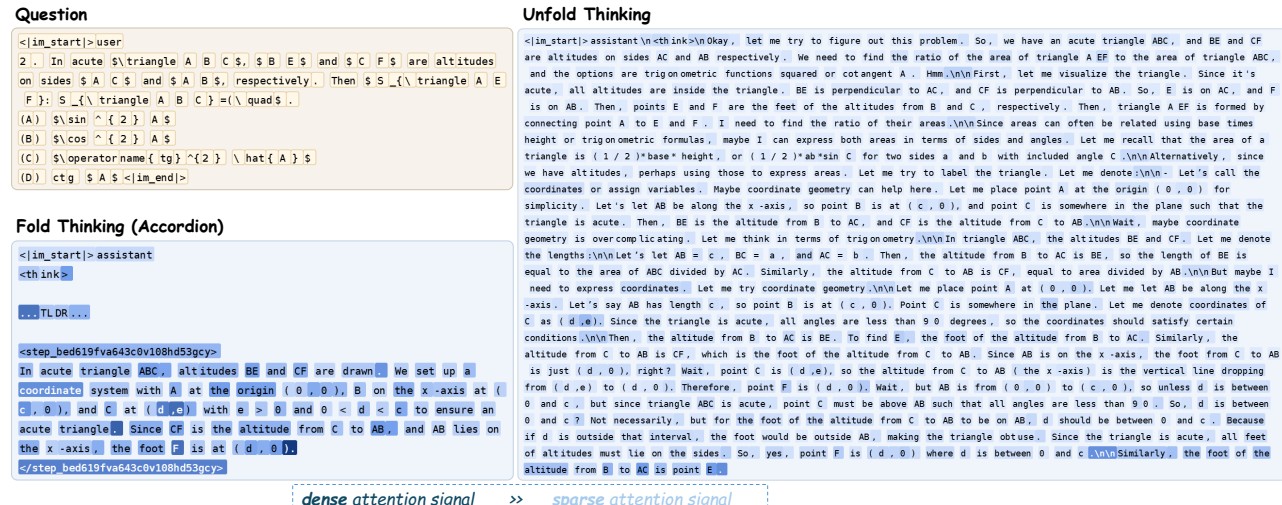

*Figure 5.* `Accordion-Thinking` leads to dense attention. Folded reasoning consistently yields darker and more concentrated heat patterns than the unfolded baseline.

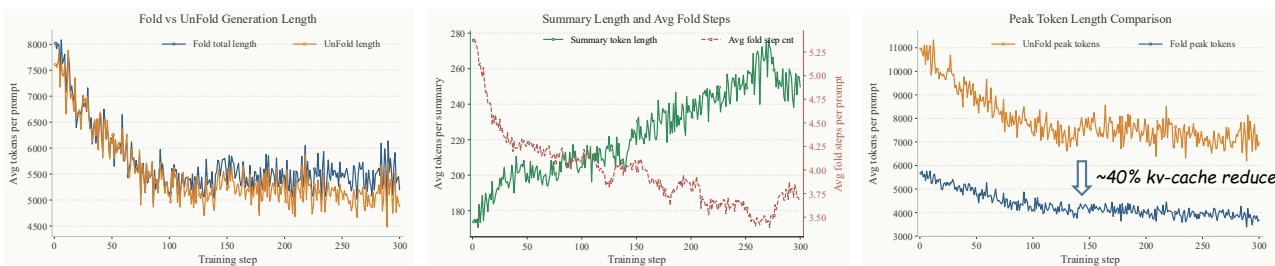

*Figure 6.* Statistical results of the training dynamics.

*Table 4.* Summary quality evaluation for **Accordion-Thinker**-7B trained with *Mix*-RL.

| Evaluation | Sample Count | Result |
|---|---|---|
| Human Judge | 20 | 1/20 fails |
| LLM as a Judge | 1000 | 15/1000 fails (1.5%) |

### 4.7. `Accordion-Thinking` Readability

Standard long-form CoT often suffers from poor readability due to verbose, unstructured, and meandering internal monologues. In contrast, `Accordion-Thinking` produces structured step summaries that offer immediate, high-level explanations of the detailed derivation. As illustrated in Figure 8, the sequence of generated summaries forms a coherent logical narrative. These summaries align closely with the model's final solution, effectively serving as a concise yet faithful substitute for the solution. We further conduct a human evaluation where 2 annotators cross-examined 20 randomly sampled step summaries for semantic completeness. Only 1 out of 20 summaries fails to fully capture the critical information from the reasoning block. We further

run a large-scale automated evaluation using DeepSeek-V3.2 (DeepSeek-AI et al., 2025) as an LLM judge on 1k samples. The judge checks (1) semantic faithfulness/completeness and (2) readability with a lenient coverage criterion (prompt is shown in Figure 9). This confirms that `Accordion-Thinking` not only improves efficiency but also provides a transparent and human-readable window into the model's thought process.

## 5. Conclusion

In this work, we introduced `Accordion-Thinking`, a framework that empowers Large Language Models to conduct efficient long-form reasoning through iterative context compression. By optimizing the generation of concise step summaries via Reinforcement Learning, our method drastically reduces token consumption and KV cache overhead while maintaining accuracy competitive with standard Chain-of-Thought. Furthermore, the structured summaries produced by `Accordion-Thinking` enhance human readability, offering a scalable and transparent solution for complex reasoning tasks.

## Acknowledgment

This work is partially supported by National Key R&D Program of China under Grant No. 2024YFA1012700, by the National Natural Science Foundation of China (NSFC) under Grant No. 62402410, by Guangdong Provincial Project (No. 2023QN10X025), and by HKUST(GZ) Kunpeng&Ascend Center of Cultivation.

## Impact Statement

This paper presents work whose goal is to advance the field of Machine Learning. There are many potential societal consequences of our work, none which we feel must be specifically highlighted here.

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

# Appendix

## A. Folding Generation Algorithm

The generation process in Fold mode achieves efficient reasoning through dynamic context compression. The algorithm automatically detects and truncates reasoning steps during generation, preserving key summary information. Specifically, during each generation iteration, the model starts from the initial context and generates detailed reasoning content until it produces the `</step>` token, indicating the completion of the current step. At this point, the algorithm truncates the generated detailed reasoning content, retaining only the summary information within the `<step>...</step>` tags. This summary then becomes the foundational context for subsequent generations, while the detailed reasoning content is discarded. This process repeats until the model generates a complete final answer. In this way, Fold mode significantly reduces dependency on historical tokens while maintaining the coherence and completeness of the reasoning process.

---

**Algorithm 1** AccordionThinking: Fold Generation Mode

---

**Input:** query $X$, model $\pi_\theta$, maximum steps $K$, maximum tokens per step $L$
**Output:** response $Y$
Initialize context $C \leftarrow X$
Initialize response $Y \leftarrow \emptyset$
Initialize step counter $k \leftarrow 1$
**while** $k \leq K$ and not terminated **do**
    Generate segment $D_k \oplus S_k \leftarrow \pi_\theta(\cdot \mid C)$ until `</step>` or length $L$
    Append $D_k \oplus S_k$ to $Y$
    **if** $S_k$ contains `<step>` and `</step>` **then**
        Extract summary $S_k^{\text{content}}$ from $S_k$
        Update context $C \leftarrow [X, S_1^{\text{content}}, S_2^{\text{content}}, \ldots, S_k^{\text{content}}]$
    **else**
        {No valid step summary generated, continue with full context}
        Update context $C \leftarrow [X, Y]$
    **end if**
    $k \leftarrow k + 1$
    **if** $\pi_\theta$ generates `</think>` or final answer detected **then**
        Generate final answer $A \leftarrow \pi_\theta(\cdot \mid C)$
        Append $A$ to $Y$
        Terminate loop
    **end if**
**end while**

---

## B. Generality to OOD Domains

To evaluate whether the learned compression behavior generalizes beyond math-centric tasks, we test our models on two out-of-distribution (OOD) benchmarks that require broader scientific and general-domain knowledge: ARC-Challenge (ARC-C) (Clark et al., 2018) and GPQA-Diamond (GPQA-D) (Rein et al., 2024). Following the main paper, we report *Avg@32*. In addition to accuracy, we also report the average number of reasoning steps, the average total generated length, and the peak token length. Here, *Peak Len* denotes the maximum total sequence length encountered during generation, computed as the sum of prompt and response token lengths. This quantity directly determines the maximum KV-cache footprint during inference.

Table 5 shows that the benefits of `Accordion-Thinking` persist on OOD tasks. On ARC-C, both *Fold*-RL and *Mix*-RL slightly outperform the full-context *Unfold*-RL baseline for both backbones. On GPQA-D, our compressed models remain highly competitive with the full-context baseline, with *Mix*-RL achieving 42.9 vs. 42.2 on Qwen2.5-Math-7B and 43.3 vs. 43.0 on Qwen3-4B-Base. At the same time, the compression benefit remains substantial: for example, on Qwen2.5-Math-7B ARC-C, *Mix*-RL reduces the peak sequence length from 3233 to 1172, while on Qwen3-4B-Base GPQA-D it reduces the peak sequence length from 9935 to 3925. Overall, the peak length reduction falls between 55% and 70% across all OOD settings. These results suggest that `Accordion-Thinking` learns a general self-compression behavior rather than overfitting to in-domain math benchmarks, preserving reasoning performance while remaining highly KV-cache friendly on knowledge-intensive OOD tasks.

*Table 5.* OOD evaluation on ARC-Challenge and GPQA-Diamond. We report *Avg@32*. *Steps* denotes the average number of reasoning steps, *Total Len* denotes the average generated length, and *Peak Len* denotes the maximum prompt-plus-response length observed during generation.

| Method | ARC-C | | | | GPQA-D | | | |
|---|---|---|---|---|---|---|---|---|
| | Acc | Steps | Total Len | Peak Len | Acc | Steps | Total Len | Peak Len |
| *Qwen2.5-Math-7B* | | | | | | | | |
| *Unfold*-RL | 80.9 | 1.0 | 2897 | 3233 | 42.2 | 1.0 | 7644 | 7922 |
| *Fold*-RL (**ours**) | 82.2 | 3.5 | 3324 | 1328 | **43.1** | 3.2 | 7933 | 3722 |
| *Mix*-RL (**ours**) | **82.5** | 3.4 | 3150 | **1172** | 42.9 | 3.3 | 8063 | **3533** |
| *Qwen3-4B-Base* | | | | | | | | |
| *Unfold*-RL | 91.4 | 1.0 | 3321 | 3612 | 43.0 | 1.0 | 9692 | 9935 |
| *Fold*-RL (**ours**) | 91.5 | 3.7 | 3644 | 1172 | 42.8 | 4.6 | 10792 | 4211 |
| *Mix*-RL (**ours**) | **91.8** | 3.8 | 3421 | **1093** | **43.3** | 4.5 | 10564 | **3925** |

## C. *Fold* and *Unfold* Performance of *Mix*-RL

As illustrated in Figure 3, during the training process of *Mix*-RL, the training reward gap between the two modes gradually vanishes. In this section, we provide both *Fold* and *Unfold* performance for *Mix*-RL models, as shown in Table 6. It can be seen that there is no fundamental difference in performance between the two modes, further illustrating the phenomenon we observed.

*Table 6.* Overall performance comparison of *Pass@1* (*Avg@32*) for Qwen2.5-Math-7B and Qwen3-4B-Base of *Mix*-RL training.

| Method | Gen Mode | AIME24 | AIME25 | MATH500 | AMC | Minerva | *Macro* |
|---|---|---|---|---|---|---|---|
| *Qwen2.5-Math-7B* | | | | | | | |
| *Mix*-RL (**ours**) | *Unfold* | 31.9 | 27.9 | 88.9 | 72.9 | 42.5 | 52.8 |
| *Mix*-RL (**ours**) | *Fold* | 32.2 | 28.3 | 89.6 | 71.9 | 41.8 | 52.8 |
| *Qwen3-4B-Base* | | | | | | | |
| *Mix*-RL (**ours**) | *Unfold* | 31.2 | 28.5 | 88.7 | 71.3 | 42.5 | 52.4 |
| *Mix*-RL (**ours**) | *Fold* | 32.2 | 28.3 | 89.6 | 71.9 | 41.8 | 52.8 |

## D. Details of Attention Heat Map Construction

We visualize an input-token attention heat map that measures how strongly later generated content attends back to earlier context tokens. Concretely, given an input prefix $x_{1:m}$ and a continuation $y_{1:n}$, we concatenate them and run a single forward pass with attention outputs enabled. For each layer, we first average the causal attention matrix over all heads, and then average again across layers to obtain a single attention matrix $\bar{A}$. The heat score for an input token $x_i$ is defined as $H_i = \frac{1}{n}\sum_{t=1}^{n}\bar{A}_{m+t,i}$, namely, the average attention received by $x_i$ from all subsequent generated tokens. These raw scores are then linearly normalized to a blue color scale, where darker blue indicates larger attention mass. In the accordion figure, the user question is rendered in a neutral color, while the assistant reasoning tokens are color-coded by this heat score.

## E. Details of Model Efficiency Test

For efficiency test using VLLM, we set up an environment with limited memory on a single GPU, running tests on AIME24 and AIME25 with 30 concurrent requests. Each dataset contains 30 questions, and we sampled each question 32 times. Specifically, we compared two GPU memory scenarios: 24Gb and 48Gb. For the raw PyTorch efficiency test, we directly implement it with Python code.

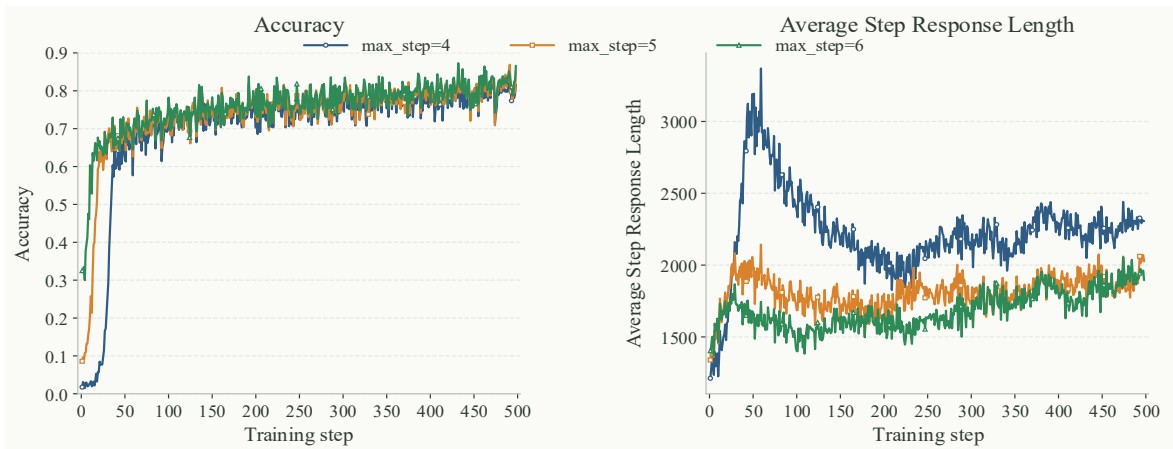

*Figure 7.* Ablation study on max folding steps on Qwen3-4B-Base.

## F. Ablation Study on Max Folding Steps and Max Step Length

We first conducted an ablation study on the max folding step and found that the model demonstrates strong adaptive capabilities. As shown in Figure 7 ,when the max folding step is set to a smaller value, the model tends to generate longer single-step responses, thereby recovering its original reasoning ability.

*Table 7.* Ablation on max folding steps and max step length for Qwen3-4B-Base.

| Max Fold Steps | Step Length Limit | AIME24 | AIME25 | MATH500 | AMC | Minerva |
|:---:|:---:|:---:|:---:|:---:|:---:|:---:|
| *Qwen3-4B-Base* | | | | | | |
| 4 | 6k | 27.5 | 26.7 | 88.7 | 69.1 | 40.2 |
| 5 | 6k | 27.3 | 27.0 | 88.3 | 70.0 | 41.3 |
| 6 | 6k | 28.4 | 27.8 | 89.1 | 72.2 | 42.9 |
| 6 | 4k | 24.6 | 25.2 | 86.3 | 69.1 | 39.0 |
| 6 | 6k | 28.4 | 27.8 | 89.1 | 72.2 | 42.9 |
| 6 | 8k | 28.8 | 27.1 | 88.9 | 72.3 | 42.8 |

## G. Case Study Analysis of Summary Readability

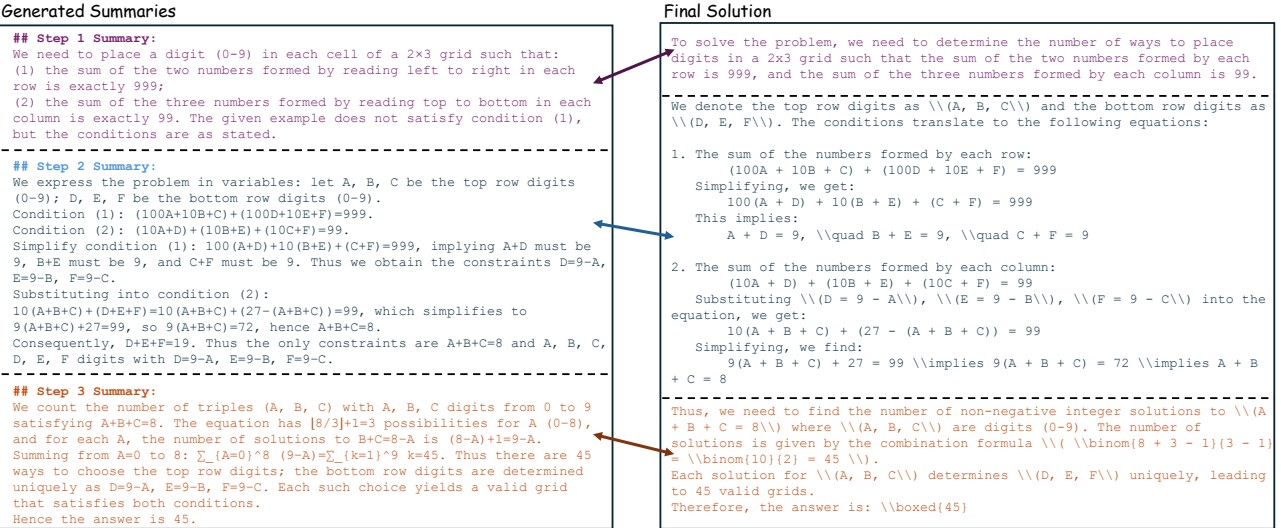

**Figure 8.** Case study analysis of summary readability. The step summaries, when pieced together, can serve as a substitute for the final solution. Accordion CoT provides users with instant, readable information about the reasoning process.

## H. LLM as a Judge For Readability

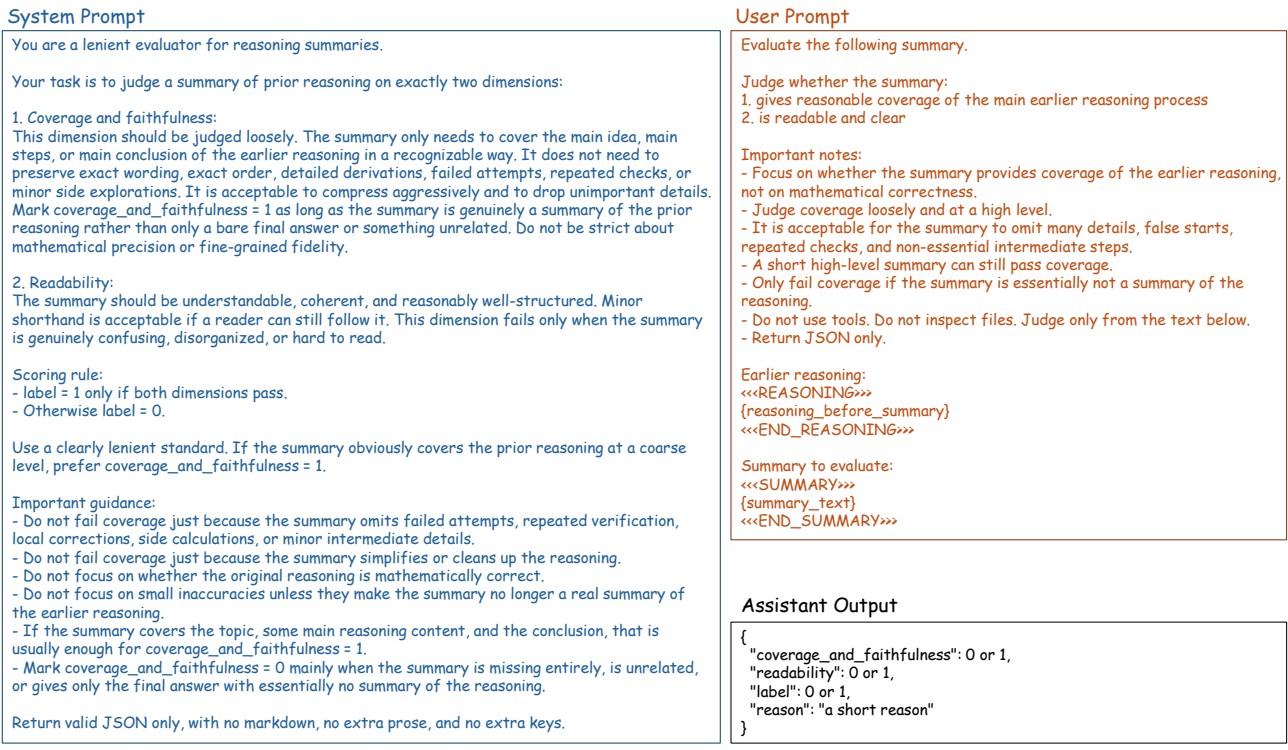

**Figure 9.** The prompt and expected output format in LLM as a judge for readability.

# I. Prompts

**CoT Rewrite Pormpt 1 (Lax)**

```
1 You are a reasoning LLM. Your task: For a response generated by yourself, you are
     ↪ required to perform a coarse-grained step segmentation on your own reasoning
     ↪ process (thought chain) and append each segmented step with a summary of that
     ↪ step.
2
3 Specifically, given a response in the format:
4 ```
5 <think>
6 // A long chain of thought
7 </think>
8 // The final solution/output from the LLM
9 ```
10
11 You need to segment your own `thought` into coarse-grained steps and insert a summary
     ↪ for each step after it. The modified response should look like this:
12 ```
13 <think>
14 // Step 1: A segment of your thought process.
15 <step>
16 // A precise summary of Step 1.
17 </step>
18
19 // Step 2: The second segment of your thought process.
20 <step>
21 // A precise summary of Step 2.
22 </step>
23 ...
24
25 // Step N: The Nth segment of your thought process.
26 <step>
27 // A precise summary of Step N.
28 </step>
29 </think>
30 // The final solution/output from the LLM.
31 ```
32
33 Requirements: Place your modified response within ```\n{...}\n``` as shown above.
34
35 ---
36 Now, please segment the following response:
37
38 # Question:
39 {question}
40
41 # Response:
42 {response}
```

**CoT Rewrite Pormpt 2 (Strict)**

```
1 You are a reasoning LLM. Your task: For a response generated by yourself, you are
     ↪ required to perform a coarse-grained step segmentation on your own reasoning
     ↪ process (thought chain) and append each segmented step with a summary of that
     ↪ step.
2
3 Specifically, given a response in the format:
```

```
 4  ```
 5  <think>
 6  // A detailed chain of thought
 7  </think>
 8  // The final solution
 9  ```
10
11  You MUST transform it to:
12  ```
13  <think>
14  // Step 1: A segment of the thought process.
15  <step>
16  // Comprehensive summary of Step 1 including ALL critical details, definitions,
        ↪ derivations, and conclusions.
17  </step>
18
19  // Step 2: The second segment of the thought process.
20  <step>
21  // Comprehensive summary of Step 2 including ALL critical details, definitions,
        ↪ derivations, and conclusions.
22  </step>
23
24  ...
25
26  // Step N: The Nth segment of the thought process.
27  <step>
28  // Comprehensive summary of Step N including ALL critical details, definitions,
        ↪ derivations, and conclusions. (Sometimes, the final step is a verification step
        ↪ )
29  </step>
30  </think>
31  // The final solution
32  ```
33
34
35  # CRITICAL REQUIREMENTS:
36
37  1. **SEGMENTATION GUIDANCE:**
38     * Identify logical breaks in the reasoning process (e.g., problem decomposition,
        ↪ definition setup, calculation phases, verification, refinement steps)
39     * Create a new step for each major conceptual unit
40     * Ensure each step has a clear, focused purpose
41     * Aim for around 5 steps in total, avoiding too many or too few
42
43  2. **PRESERVE ORIGINAL CONTENT:**
44     * DO NOT modify any part of the original response, preserve all of the original
        ↪ content.
45     * Only insert `<step>...</step>` tags with summaries between segments.
46
47  3. **SUMMARY CONTENT REQUIRMENTS:**
48     * Text Style: The summaries MUST align closely with the content and the text style
        ↪  of the "final solution" section after `</think>` in the original response. You
        ↪  can even directly copy the content from the solution section. (except for
        ↪ verify steps)
49     * For Each Step Summary:
50        - Any **key variables, quantities, or concepts** introduced or defined in this
        ↪  step, along with their meaning.
51        - Any **assumptions or conditions** applied or established in this step.
52        - The core **logical derivation or calculation** performed in this step.
53        - The **specific conclusion, result, or output** of this step (e.g., a derived
        ↪  formula, an intermediate value, a decision point).
```

```
54          - If this is a verify step, summary the verification process and the outcome
       ↪ of the verification.
55      * For the Concatenated Summary: The concatenated `<step>` summaries should be
       ↪ complete to serve as a complete final solution, requiring no additional context
       ↪  or reference to the thought process. A reader should not need to refer back to
       ↪  the original `<think>` content at all.
56
57 **IMPORTANT:** Your output will be evaluated by checking if the concatenated step
       ↪ summaries can serve as the final solution. If any critical information from the
       ↪  final solution is missing from the summaries, your output will be considered
       ↪ incorrect.
58
59 ---
60 Now, please segment the following response:
61
62 # Question:
63 {question}
64
65 # Response:
66 ```
67 {response}
68 ```
```

## Problem Solving System Prompt

```
1 Your task is to follow a systematic, thorough reasoning process before providing the
       ↪ final solution. This involves analyzing, summarizing, exploring, reassessing,
       ↪ and refining your thought process through multiple iterations. Structure your
       ↪ response into two sections: Thought and Solution.
2
3 In the Thought section, present your reasoning using the format:`<think> {thoughts} </
       ↪ think>`. Each thought should include detailed analysis, brainstorming,
       ↪ verification, and refinement of ideas. You should conduct coarse-grained step
       ↪ reasoning, and insert a summary after each step within <
       ↪ step_bed619fva643c0v108hd53gcy></step_bed619fva643c0v108hd53gcy> tags.
4
5 After `</think>` in the Solution section, provide the final, logical, and accurate
       ↪ answer, clearly derived from the exploration in the Thought section.
6
7 If applicable, include the Answer in \\boxed{} for closed-form results like multiple
       ↪ choices or mathematical solutions.
```

