# OpenReview forum: "Accordion-Thinking: Self-Regulated Step Summaries for Efficient and Readable LLM Reasoning"
_ICML.cc/2026/Conference — ICML 2026 regular_

### Official Review · Reviewer_Lid5 · 2026-03-12

**Soundness:** 3
**Presentation:** 3
**Significance:** 2
**Originality:** 3
**Overall Recommendation:** 5
**Confidence:** 3

**Summary:**

This work strives to examine an important concept in efficient large language model reasoning of reducing the computational cost of long Chain-of-Thought reasoning while preserving solution quality. The article's principal contribution consists of the proposed Accordion-Thinking framework, which enables a model to periodically summarize intermediate reasoning steps and discard earlier detailed reasoning tokens during inference. The framework structures reasoning into alternating segments of detailed reasoning and concise step summaries. During a compressed "Fold" inference mode, only the summaries are retained in the context while detailed reasoning is removed from the KV cache, which reduces memory usage and attention complexity. The method is trained using a combination of supervised fine tuning on synthetically structured reasoning traces and reinforcement learning that rewards correct answers under compressed context conditions. The training encourages the model to encode critical reasoning state into summaries so that later steps can rely on them.

Experiments are conducted on mathematical reasoning benchmarks using Qwen based models. The results show that models trained with the proposed method can match the accuracy of full Chain-of-Thought reasoning while operating in the compressed Fold mode. The paper also reports improved throughput under constrained GPU memory conditions and presents qualitative and human evaluation results suggesting that the generated summaries are readable and capture key reasoning steps.

**Compliance With Llm Reviewing Policy:**

Affirmed.

**Final Justification:**

The author response addresses my concerns. I increased my score accordingly.

**Key Questions For Authors:**

How does Accordion-Thinking perform on reasoning tasks outside of mathematics, such as logical reasoning, planning, or coding tasks Evidence across more domains is the clearest way to strengthen the paper.

How sensitive is the method to the size and quality of the synthetic Accordion dataset used during the cold start stage? Clarifying this would help determine whether the approach scales to broader tasks or relies heavily on carefully constructed training data.

Can the authors provide more detailed measurements of memory usage, latency, and batch scaling during inference in Fold mode compared to standard Chain-of-Thought? Such analysis would help quantify the practical deployment benefits more precisely.

**Limitations:**

yes

**Strengths And Weaknesses:**

**Soundness**

Strengths
- The paper clearly formulates the Fold and Unfold reasoning modes and provides a concrete algorithmic description of the context pruning mechanism.
- The training pipeline combines supervised fine tuning and reinforcement learning with verifiable rewards, which is a reasonable approach for training reasoning models.
- Experiments evaluate multiple models and several mathematical reasoning benchmarks, which helps support the empirical claims about accuracy and efficiency.
- The reinforcement learning setup uses binary rewards based on answer correctness, which is transparent and easy to interpret.


Weaknesses
- The synthetic dataset used to train the summarization capability is relatively small, and it is unclear how sensitive the method is to the size or diversity of this dataset.
- The human evaluation for summary quality is limited to a very small sample, which weakens the strength of claims about readability or semantic faithfulness.



**Presentation**

Strengths
- The paper is generally well structured (and I really enjoyed the figures), with a clear separation between the method, data synthesis pipeline, and reinforcement learning strategy.
- The distinction between Fold and Unfold modes is clearly defined and illustrated conceptually.
- The experimental tables provide direct comparisons between training strategies, which helps clarify the contribution of the proposed method.


Weaknesses
- No notable weaknesses. Small thing: it might be nice to move figure 1 to earlier in the paper since it very nicely depicts the method - e.g., page 2 or 3.



**Significance**

Strengths
- The work addresses an important practical limitation of long Chain-of-Thought reasoning, namely the growth of context length and KV cache during extended reasoning.
- If the approach generalizes beyond the presented tasks, it could improve the efficiency of long reasoning trajectories in deployment settings.
- The proposed method may also improve interpretability through structured summaries of reasoning steps.


Weaknesses
- The experiments focus primarily on mathematical reasoning benchmarks, so the generality of the approach to other reasoning domains remains unclear.
- The reported gains are mainly in efficiency and compression rather than large improvements in reasoning accuracy.

**Originality**

Strengths
- The idea of learning to summarize reasoning steps and then discarding earlier detailed tokens is a novel way to integrate compression into the reasoning process itself.
- The combination of structured step summaries with reinforcement learning that forces reasoning under compressed context provides an interesting training signal.

Weaknesses
- The work builds on several existing lines of research on Chain-of-Thought reasoning, reasoning compression, and token management, so the conceptual novelty is primarily in how these ideas are combined rather than in a completely new architecture.

---

> ### Author Rebuttal · Authors · 2026-03-30
>
> Thank you for the constructive feedback and for highlighting the soundness and originality of our work. Below, we address your specific weaknesses and key questions in detail, providing additional experiments to further strengthen the paper.
>
> ---
> ### **1. Generality to Non-Math Domains (Question 1 & Weakness 4)**
> You correctly point out that demonstrating our method outside of mathematical reasoning significantly strengthens the paper. To evaluate how our model handles tasks requiring broader domain knowledge and logical reasoning, we tested our model on Out-of-Distribution (OOD) benchmarks including GPQA-Diamond and ARC-Challenge.
>
> |Model|Method|ARC-C(Acc/PeakLen)|GPQA-D(Acc/PeakLen)|
> |-|-|-|-|
> |Qwen2.5-Math-7B|UnFold-RL(ColdStart+RL)|80.9/3233|42.2/7922|
> ||Fold-RL|82.2/1328|43.1/3722|
> ||Mix-RL(Ours)|**82.5**/**1172**|**42.9**/**3533**|
> |Qwen3-4B-Base|UnFold-RL(ColdStart+RL)|91.4/3612|43.0/9935|
> ||Fold-RL|91.5/1172|42.8/4211|
> ||Mix-RL(Ours)|**91.8**/**1093**|**43.3**/**3925**|
>
> We report Avg@32 performance. Peak Token refers to the maximum sequence length governing the KV-cache bottleneck
> Despite being OOD and requiring different reasoning paradigms, Accordion-Thinking gracefully adapts its dynamic context management. It maintains high accuracy while **reducing the peak token footprint by roughly 50%-60%**, proving that the self-regulated folding mechanism generalizes well beyond mathematics.
>
> ---
> ### **2. Sensitivity to the SFT Dataset (Question 2 & Weakness 1)**
> To systematically evaluate the sensitivity of the Cold-Start stage, we conducted additional experiments on both the **quantity** and the **quality** of the SFT dataset.
>
> **Sensitivity to Data Quantity:** We varied the size of the synthetic SFT dataset to observe its impact on the performance in `Fold` mode:
>
> | Data Size |AIME24|MATH-500|AMC|
> |-|-|-|-|
> |4,000|17.9|75.8|54.2|
> |8,000|21.3|79.8|60.2|
> |14,653|**23.0**|**82.3**|**62.4**|
>
> **Sensitivity to Data Quality:** In Figure 2 of our paer, we already show that strict prompt synthesize better quality data and improve the cold-start performance. Furthermore, we replaced the SFT data source with **our** own RL-trained model (`Accordion-Thinker-7B` with Mix-RL) and re-evaluated the performance of the base models. Data generated by our own model yields significantly better performance than data from DS-V3.2. We attribute this to the fact that synthetic data relies on an artificial, prompted rewriting rule to segment the steps. In contrast, data generated by Accordion-Thinker represent the **native, self-regulated CoT**. It is closer to the LLM policy distribution, making it easier for the base model to learn.
>
> |Model|SFTData|AIME24|MATH500|AMC|
> |-|-|-|-|-|
> |Qwen2.5-Math-7B|DS-V3.2-Syn-14k|23.0|82.3|62.4|
> ||MixRL-Gen-7k|**29.5**|**86.3**|**66.6**|
> |Qwen3-4B-Base|DS-V3.2-Syn-14k|19.2|79.2|57.3|
> ||MixRL-Gen-7k|**22.5**|**84.7**|**59.7**|
>
> ---
> ### **3. Detailed Measurements for Deployment (Question 3)**
> We detail our efficiency gains across two distinct inference settings:
>
> * **Raw PyTorch:** Tested directly on a 48GB GPU without concurrency. Vanilla CoT latency grows quadratically, whereas Accordion-Thinking bounds the active context, maintaining a low, "saw-tooth" latency profile.
> * **vLLM (Practical Batch Scaling):** Tested under strict 24/48GB memory limits with 256 concurrent requests. In vLLM, maximum concurrent is bottlenecked by KV-cache capacity. Vanilla CoT's linear context growth rapidly exhausts this cache, forcing the engine to halt new request dispatches. By drastically reducing the **Peak Token** footprint, Accordion-Thinking fits significantly more concurrent requests into the exact same memory budget, directly yielding the **~3x** throughput improvement.
>
> Peak Token (k) Statistical:
> |Model|Method|AIME24|AIME25|Math500|AMC|Minerva|
> |-|-|-|-|-|-|-|
> |Qwen2.5-Math-7B|ColdStart+RL|10.3|11.3|4.9|7.2|6.9|
> ||Fold-RL|4.4|4.5|2.4|3.1|2.9|
> ||Mix-RL|4.4|4.4|2.1|3.0|2.9|
> ||
> |Qwen3-4B-Base|ColdStart+RL|12.3|12.9|6.9|10.3|9.4|
> ||Fold-RL|5.9|5.5|2.8|4.0|3.7|
> ||Mix-RL|5.7|5.4|2.7|4.1|3.6|
>
> ---
> ### **4. Summary Quality and Sample Size (Weakness 2)**
> To address your concern regarding the limited sample size, we conducted a comprehensive automated evaluation using LLM as a judge. We randomly extracted 1k reasoning-summary pairs generated by Accordion-Thinker-7B(mixRL). We then prompted DS-V3.2 to evaluate each pair based on two criteria: (1) Semantic Faithfulness and Completeness: Does the summary accurately and sufficiently cover the critical reasoning content of the preceding step? and (2) Readability: Is the summary coherent, structured, and easy to follow? (The prompt is shown in [[Rebuttal-Fig4]](https://anonymous.4open.science/r/Accordion-ICML-Rebuttal-3D5D/Rebuttal-Fig4.pdf)).
>
> Only 15 summaries (**1.5%**) are flagged as negative by the judge. This indicates that Accordion-Thinking consistently produces highly readable and semantically faithful reasoning traces.

---

> > ### Author Rebuttal · Reviewer_Lid5 · 2026-04-04
> >
> > All concerns have been addressed. I appreciate the authors outlining new results above. I will update my score accordingly.

---

> > > ### Author Response · Authors · 2026-04-05
> > >
> > > Thank you for the acknowledgement of our work and for your intention to raise the score! We would greatly appreciate it if you could kindly reflect this score update in the review system.
> > >
> > > We truly appreciate the time and effort you invested in reviewing our paper. We are glad our rebuttal has adequately addressed your concerns, and we will ensure all newly added experiments, clarifications, and discussions are included in the final version.

---

### Official Review · Reviewer_ZeP1 · 2026-03-13

**Soundness:** 3
**Presentation:** 3
**Significance:** 2
**Originality:** 3
**Overall Recommendation:** 4
**Confidence:** 2

**Summary:**

The paper focuses on efficient LLM reasoning. It proposes Accordion-Thinking, a method designed to mitigate the inference time bottleneck of long Cot. It can dynamically discard the previous detailed reasoning steps, generate a summary, and then continue the inference relying solely on the summary. The paper uses the high-quality SFT and dynamic context-truncation RL (based on GRPO), and mixed-mode training (Mix-RL) to achieve this. Lastly, the method achieves an approximately 3x throughput increase without sacrificing accuracy on mathematical reasoning benchmarks.

**Compliance With Llm Reviewing Policy:**

Affirmed.

**Final Justification:**

The rebuttal properly addresses my concerns. I maintain my score.

**Key Questions For Authors:**

1. The paper assumes a maximum of $ N=6$ steps and 6144 tokens per step. However, have you tried the big reasoning model have longer steps to solve more complicated problems?

2. Is there a risk that the loss summarization of the early states could lead to a compounding error effect, resulting in an irreversible collapse of the model's logic?

3. Tiny question: Have you tried GPQA diamond, which may need more steps?

**Limitations:**

yes

**Strengths And Weaknesses:**

Strengths:
1. The paper shows a significant shift from the rule-based truncation to the model-native, learnable capability in the long-context compression domain. It is intuitively grounded.
2. The paper characterizes the "Gap-Vanishing" phenomenon, which validates the proposed method. It offers valuable insights into the information compression limits of LLMs.
3. The practical impact is also good. It significantly enhances the throughput gains. And the step summaries help people to understand the model/s reasoning logic.

Weakness:
1. See the questions.

---

> ### Author Rebuttal · Authors · 2026-03-30
>
> Thank you for your thoughtful review and for highlighting the potential of our method on more complex reasoning tasks. We appreciate your constructive feedback and address your questions in detail below.
>
> ---
> ### **1. Exploring Longer Steps and Token Limits**
> To investigate whether expanding the reasoning constraints helps the model solve more complicated problems, we conducted two sets of ablation studies on our `Mix-RL` model.
>
> First, we fixed the maximum tokens per step ($L=6144$) and varied the maximum number of steps ($N \in \{4, 5, 6, 7\}$). We also show the training dynamic of this ablation study in [[Rebuttal-Fig2]](https://anonymous.4open.science/r/Accordion-ICML-Rebuttal-3D5D/Rebuttal-Fig2.pdf).
> Second, we fixed the number of steps ($N=6$) and varied the maximum step length ($L \in \{4096, 6144, 8192\}$). The performance of Qwen3-4B-Base with FoldRL is shown below:
>
> **Table A: Varying Maximum Steps ($N$) with fixed $L=6144$**
> |Max Steps|AIME24|AIME25|MATH500|AMC|Minerva|
> |-|-|-|-|-|-|
> |4|27.5|26.7|88.7|69.1|40.2|
> |5|27.3|27.0|88.3|70.0|41.3|
> |6(Default)|28.4|**27.8**|**89.1**|72.2|**42.9**|
> |7|**28.8**|27.5|88.8|**72.5**|42.6|
>
>
> **Table B: Varying Maximum Step Length ($L$) with fixed $N=6$**
> |Max Tokens|AIME24|AIME25|MATH500|AMC|Minerva|
> |-|-|-|-|-|-|
> |4096|24.6|25.2|86.3|69.1|39.0|
> |6144(Default)|28.4|**27.8**|**89.1**|72.2|**42.9**|
> |8192|**28.8**|27.1|88.9|**72.3**|42.8|
>
> As the results indicate, increasing steps or token limits improves performance up to a point, but gains diminish beyond certain steps and tokens length, further increases yield only marginal or inconsistent improvements. This suggests the default configuration strikes an effective balance between reasoning capacity and efficiency.
>
> ---
> ### **2. Mitigating Compounding Errors via Reinforcement Learning**
> You raise a very insightful point. There is indeed a high risk of compounding errors if early summaries are lossy. In fact, we observed exactly this irreversible logic collapse during our early Supervised Fine-Tuning (SFT) stage. As shown in Table 1 of the main paper, the "Cold-Start" SFT model suffers a significant performance drop when switching from `Unfold` to `Fold` mode (e.g., dropping from 48.5% to 45.7% on Qwen2.5-Math-7B), precisely because of this compounding error effect.
>
> However, **our Reinforcement Learning (RL) pipeline is fundamentally designed to prevent this.** During RL training in `Fold` mode, the model is strictly forced to rely on its self-generated past summaries. If it drops critical information or propagates errors early on, it will fail to derive the final answer and receive a reward of 0. This hard constraint acts as a strong penalty against lossy summarization.
>
> Consequently, the RL process forces the model to learn **compression**, encoding all essential mathematical states and logic into the summary. Our "Gap-Vanishing" phenomenon (Figure 3) empirically demonstrates that as RL progresses, the model overcomes this compounding error, allowing the `Fold` mode to fully match the accuracy of the full-context `Unfold` mode.
>
> ---
> ### **3. Evaluation on GPQA Diamond (OOD Generalization)**
> Thank you for the excellent suggestion. To evaluate how our model handles tasks that require more extensive domain knowledge, we tested our model on Out-of-Distribution (OOD) benchmarks: GPQA-Diamond and ARC-Challenge. (We report avg@32 performance)
>
> | Model | Method | ARC-C (Acc / Steps / Total Len / Peak Len) | GPQA-D (Acc / Steps / Total Len / Peak Len)|
> |-|-|-|-|
> |Qwen2.5-Math-7B|UnFold-RL(ColdStart+RL)|80.9 / 1.0 / 2897 / 3233 |42.2 / 1.0 / 7644 / 7922|
> ||Fold-RL|82.2 / 3.5 / 3324 / 1328|43.1 / 3.2 / 7933 / 3722|
> ||Mix-RL(Ours)|**82.5** / 3.4 / 3150 / **1172**|**42.9** / 3.3 / 8063 / **3533**|
> ||
> |Qwen3-4B-Base|UnFold-RL(ColdStart+RL)|91.4 / 1.0 / 3321 / 3612|43.0 / 1.0 / 9692 / 9935|
> ||Fold-RL|91.5 / 3.7 / 3644 / 1172|42.8 / 4.6 / 10792 / 4211|
> ||Mix-RL(Ours)|**91.8** / 3.8 / 3421 / **1093**|**43.3** / 4.5 / 10564 / **3925**|
>
> Peak token refers to the maximum total sequence length, calculated as the sum of prompt and response token lengths. This key metric directly governs the maximum size of the KV cache required, serving as a fundamental constraint on the model’s GPU memory consumption and runtime performance. **Our method reduces the peak token by 60% while ensuring OOD task performance, which is extremely friendly to the KV cache.**
>
> Consistent with your intuition, solving GPQA Diamond indeed triggers the model to utilize more reasoning steps (average of 3.3/4.5 steps for 7B/4B model) and longer response lengths compared to standard math benchmarks.
> Despite being OOD, Accordion-Thinking gracefully adapts its dynamic context management to these complex problems, maintaining high accuracy while preserving the efficiency benefits of context folding. We will include these results in the appendix of the final version.

---

> > ### Author Rebuttal · Reviewer_ZeP1 · 2026-03-31
> >
> > The rebuttal properly addresses my concerns. I maintain my score.

---

> > > ### Author Response · Authors · 2026-04-03
> > >
> > > Dear Reviewer ZeP1,
> > >
> > > Thank you very much for your time and for acknowledging that your concerns have been fully resolved. We truly appreciate your thoughtful feedback and are glad to hear that our rebuttal has addressed the key questions you raised.
> > >
> > > Given that all your concerns are now resolved, would you kindly consider revising your overall recommendation or confidence score to better reflect the improved clarity and technical validity of our work? We believe that with these clarifications, the paper may be better positioned relative to your original assessment.
> > >
> > > Thank you again for your constructive review and valuable suggestions. If you have other questions, we are willing to conduct further discussions.
> > >
> > > Best regards,
> > >
> > > The Authors 20603

---

### Official Review · Reviewer_Q1Kd · 2026-03-13

**Soundness:** 3
**Presentation:** 2
**Significance:** 2
**Originality:** 2
**Overall Recommendation:** 4
**Confidence:** 4

**Summary:**

This paper proposes Accordion-Thinking, a reasoning framework in which the model alternates between detailed reasoning steps and concise step summaries. In Fold mode, once a step summary is produced, the detailed reasoning for that step is removed from context and future tokens must rely only on the input and past summaries. The method first uses a synthetic SFT pipeline to teach this structured format, and then applies RL in Unfold, Fold, and mixed training modes. The main claim is a “gap-vanishing” effect: although Fold mode initially underperforms full-context Unfold mode, the gap narrows during RL training, suggesting that the model learns to preserve essential information in the summaries. Experiments on one math training dataset and evaluated on several test datasets show that Fold-RL and Mix-RL can approach Unfold-mode performance while improving efficiency under memory constraints.

**Compliance With Llm Reviewing Policy:**

Affirmed.

**Final Justification:**

The authors' rebuttal initially misunderstood some of my concerns, vis-a-vis the robustness of the fold vs unfold degradation with standard SFT/RL. However, their final response addresses most of my concerns and shows that the observed degradation in performance without specialized RL training is a consistent and robust trend, which cannot be simply explained by evaluation noise. The added Delethink baseline also strengthens the contribution significantly.

After re-evaluating the paper in light of the added clarifications and robust results, I have updated my score accordingly.

**Key Questions For Authors:**

- Could the authors clarify what exactly is the mixed RL method? The description that both modes are executed “in a single training step” and then “updated sequentially” is a bit confusing. My understanding was that some training steps use fold mode and some use unfold mode so that the model learns both capabilities.
- Could the authors explain how they picked the values of $N$ and $L$? Did they try other values and find this to work best? How sensitive is Accordion-Thinking to these hyperparameters?
- Seeing existing works like Delethink that have a similar RL training pipeline except they drop tokens from the history instead of summarizing it - is there a specific reason why summarizing is preferable over deleting tokens? Both are fundamentally lossy, and Delethink seems to work well from their results. An empirical comparison would certainly help here.

**Limitations:**

The paper does not discuss the limitations of the work. Some points to discuss here include: (1) the evaluation is restricted to math benchmarks, so broader claims about long-horizon reasoning and transparent reasoning seem under-supported; (2) the main results are presented without confidence intervals or variance, which makes them less robust; (3) the dependence on a synthetic teacher-rewritten SFT pipeline.

**Strengths And Weaknesses:**

### Strengths

- The paper proposes a method to reduce long reasoning compute cost without hurting performance, which is a useful and important problem.
- The throughput gains are substantial, without hurting performance on the math tasks used in the experiments.
- The authors did a careful ablation of the data generation pipeline, which is useful for understanding what factors help teach the model to follow the structure effectively.

### Weaknesses

- **Novelty claims vs similar existing work.** The core mechanism of generating a summary after each step and continuing subsequent reasoning using only the accumulated summaries is not a new paradigm. This work has substantial overlaps with prior works such as LightThinker [1] and InftyThink [2], which explicitly train the model to do this summarize-and-continue process. The pipeline used to generate the SFT data also closely mirrors the procedure used in these works, which is not sufficiently discussed. In addition, works like Delethink [3] drop tokens instead of summarizing and then use RL to train the model to use this compact state. Given this overlap, the paper overstates its novelty (for eg, in line 126) and does not clearly explain what is fundamentally new.
- **The paper is missing several relevant baselines.** The main results compare mostly against the paper’s own variants and does not include relevant prior work [1,2,3]. Notably, the paper cites papers like LightThinker and Delethink but does not include them as experimental baselines. More broadly, there are no direct external baselines from the recent efficient-reasoning literature (see works mentioned above) and it is difficult to understand the merits of Accordion-Thinking compared to current literature.
- **The empirical results do not clearly establish meaningful performance differences.** The main empirical results in Table 1 compare unfold and fold variants, but many of the reported differences are numerically very small. On small benchmarks such as AIME and AMC, a 2-3% gap amounts to roughly one question, making the claimed degradation under standard SFT/RL appear less substantial than the text suggests. Without confidence intervals, repeated runs, or any estimate of variance, it is difficult to know whether these gaps reflect a robust effect or are simply evaluation noise. While Accordion-Thinking does appear to reduce the fold-vs-unfold gap, the current results do not make it clear whether the improvement is large enough to be reliable or practically meaningful.
- **Experiments limited to one task setting.** The experiments only train models using RL on one math dataset. While I understand computational constraints, these observations (especially since the differences are small as noted above) may not translate to other tasks and domains.
- **Important design choices are insufficiently justified or ablated.** The folding schedule and summary structure seem to be key hyperparameters, since they primarily determine throughput gains. The authors set maximum number of steps $N=6$ and the length of reasoning step $L=6144$ but without justification or sensitivity analysis, this choice feels somewhat arbitrary.
- **The readability / faithfulness claims are under-evaluated.** The paper makes strong claims that the summaries are human-readable, semantically faithful, and can often substitute for final solutions, but the human evaluation consists of only 20 sampled summaries judged by 2 annotators, with no description of the evaluation rubric or methodology.
- **Minor points:**
    - Section 2.1 discusses some prior works that suggest RL training only enhances existing capabilities of the model, but a growing body of work challenges this narrative [4,5].
    - Line 75: typing error, space between “and” and “readable”
    - Line 86: typing error, GPRO
    - Line 231: probably meant RLVR instead of RLHF

*[1] Zhang, Jintian, et al. "Lightthinker: Thinking step-by-step compression." Proceedings of the 2025 Conference on Empirical Methods in Natural Language Processing. 2025.*

*[2] Yan, Yuchen, et al. "Inftythink: Breaking the length limits of long-context reasoning in large language models." arXiv preprint arXiv:2503.06692 (2025).*

*[3] Aghajohari, Milad, et al. "The markovian thinker: Architecture-agnostic linear scaling of reasoning." arXiv preprint arXiv:2510.06557 (2025).*

*[4] Liu, Mingjie, et al. "ProRL: Prolonged Reinforcement Learning Expands Reasoning Boundaries in Large Language Models." The Thirty-ninth Annual Conference on Neural Information Processing Systems, 2025.*

*[5] Zhang, Charlie, Graham Neubig, and Xiang Yue. "On the interplay of
pre-training, mid-training, and rl on reasoning language models." arXiv preprint arXiv:2512.07783 (2025).*

---

> ### Author Rebuttal · Authors · 2026-03-30
>
> Thank you for the detailed review. Below we reply concisely.
>
> ---
> ## **1. Related Work, Novelty, and Baselines**
>
> `LightThinker` and `Delethink` are already discussed in Sec. 2.2; we will add `InftyThink` in the revised version. Our novelty is not generic summarize-and-continue, but that the model **self-generates readable summaries as the reasoning state**, and this ability is learned mainly through **Fold-RL / Mix-RL**.
>
> |Work|Main idea|Our difference|
> |-|-|-|
> |LightThinker / Delethink|Compact representations or chunk deletion with externally specified compression|Accordion-Thinking uses **self-generated readable step summaries** as the carried state without fixed deletion rules.|
> |InftyThink|**Heavy SFT** on **333k** summarize-and-continue data.|(1) **Weak SFT** on **14k** data. (2) In-depth discussion of the performance degradation caused by Fold CoT, we show that **SFT on teacher written data is not enough, and Fold/Mix RL can eliminate the performance gap**.|
>
> We also added direct external baselines:
>
> |Model|Method|AIME24|AIME25|MATH500|AMC|Minerva|Macro|
> |-|-|-:|-:|-:|-:|-:|-:|
> |Q2.5-7B|Unfold-RL|32.0|26.7|89.2|71.2|42.1|52.2|
> ||H2O|25.1|20.3|82.9|61.5|34.7|44.9|
> ||LightThinker|27.2|22.5|84.4|67.7|37.0|47.8|
> ||Mix-RL|**32.2**|**28.3**|**89.6**|**71.9**|41.8|**52.8**|
> |Q3-4B|Unfold-RL|27.5|27.8|88.9|73.2|42.5|52.0|
> ||H2O|22.7|20.9|80.4|64.7|34.6|44.7|
> ||LightThinker|23.9|23.2|84.0|67.7|39.0|47.6|
> ||Mix-RL|**27.6**|**28.0**|88.6|72.8|**43.4**|**52.1**|
>
> Cold-start SFT alone is insufficient, and self-generated data is better than teacher-written data. This supports that **summary quality must align with the model's own policy; teacher-written summaries alone are not enough**.
>
> |Model|Data|AIME24|MATH500|AMC|
> |-|-|-:|-:|-:|
> |Q2.5-7B|DS-V3.2-Syn-14k|23.0|82.3|62.4|
> ||MixRL-7B-Gen-7k|**29.5**|**86.3**|**66.6**|
> |Q3-4B|DS-V3.2-Syn-14k|19.2|79.2|57.3|
> ||MixRL-7B-Gen-7k|**22.5**|**84.7**|**59.7**|
>
> ---
> ## **2. Meaningful Effect = Accuracy Preserved Under Compression**
>
> Our goal is **not** a large absolute acc gain over Unfold. Our goal is **Fold ≈ Unfold** with much smaller active context. The Fold gap is `-2.8 / -4.9` after cold-start SFT, `-1.9 / -2.8` after Unfold-RL, and becomes `+0.6 / +0.1` with Mix-RL on Q2.5-7B / Q3-4B.
>
> The efficiency side is shown directly by `PeakToken`, which determines the KV-cache bottleneck:
>
> |Model|Method|PeakToken|AvgFolds|AvgGenLen|
> |-|-|-:|-:|-:|
> |Q2.5-7B|Unfold-RL|6896|-|6544|
> ||Mix-RL(Unfold)|6523|-|6244|
> ||**Mix-RL(Fold)**|**2853**|3.57|6712|
> |Q3-4B|Unfold-RL|8512|-|8231|
> ||Mix-RL(Unfold)|8311|-|8096|
> ||**Mix-RL(Fold)**|**2974**|4.07|8532|
>
> So the meaningful result is: **Fold keeps accuracy essentially unchanged while reducing PeakToken by about 2.3x-2.9x**, which directly explains the efficiency gain. We will add repeated-run statistics / CIs in the revised version.
>
> ---
> ## **3. Beyond One Math Setting, Hyperparameters, and Readability**
>
> OOD transfer:
>
> |Model|Method|ARC-C(Acc/PeakLen)|GPQA-D(Acc/PeakLen)|
> |-|-|-|-|
> |Q2.5-7B|Unfold-RL|80.9/3233|42.2/7922|
> ||Mix-RL|**82.5/1172**|**42.9/3533**|
> |Q3-4B|Unfold-RL|91.4/3612|43.0/9935|
> ||Mix-RL|**91.8/1093**|**43.3/3925**|
>
> Sensitivity to `N` and `L` (Q3-4B):
>
> |Steps|AIME24|AIME25|MATH500|AMC|Minerva|
> |-|-:|-:|-:|-:|-:|
> |4|27.5|26.7|88.7|69.1|40.2|
> |5|27.3|27.0|88.3|70.0|41.3|
> |6(default)|28.4|**27.8**|**89.1**|72.2|**42.9**|
> |7|**28.8**|27.5|88.8|**72.5**|42.6|
>
> |MaxTokens|AIME24|AIME25|MATH500|AMC|Minerva|
> |-|-:|-:|-:|-:|-:|
> |4096|24.6|25.2|86.3|69.1|39.0|
> |6144(default)|28.4|**27.8**|**89.1**|72.2|**42.9**|
> |8192|**28.8**|27.1|88.9|**72.3**|42.8|
>
> These ablations show diminishing returns beyond the default; we therefore choose `N=6, L=6144` as a balanced setting.
>
> Readability / faithfulness:
>
> |Eval|N|Criterion|Result|
> |-|-:|-|-|
> |Human|20|semantic completeness|1/20 fails|
> |LLM as a judge|1000|faithfulness + readability|15/1000 negative (1.5%)|
>
> The prompt for LLM as a judge is shown in [[Rebuttal-Fig4]](https://anonymous.4open.science/r/Accordion-ICML-Rebuttal-3D5D/Rebuttal-Fig4.pdf)
>
> ---
> ## **4. Mixed RL, Deletion vs Summarization, and Minor Fixes**
>
> Mixed RL means that in each outer iteration we generate both **Fold** and **Unfold** rollouts, then apply their updates sequentially. It is a joint training regime, not a third inference mode.
>
> |Choice|Carry state|Consequence|
> |-|-|-|
> |Deletion|Shorter residual context|Efficient, but does not rewrite the reasoning state into a readable/reusable form.|
> |Accordion-Thinking|Model-generated summary|Efficient **and** interpretable; the model must compress critical logic into an explicit state.|
>
> [[Rebuttal-Fig3]](https://anonymous.4open.science/r/Accordion-ICML-Rebuttal-3D5D/Rebuttal-Fig3.pdf) further shows that **Fold mode makes attention more concentrated** on the compact carried state, instead of diffusing over a long historical trace.
>
> In the revised version we will also: add `InftyThink` and other related work, update Sec. 2.1, and fix typos (`and readable`, `GRPO`, `RLVR`).

---

> > ### Author Rebuttal · Reviewer_Q1Kd · 2026-04-02
> >
> > Thank you for the detailed rebuttal. The added baselines, OOD results, and sensitivity analysis are helpful, but my main concerns are only partially resolved.
> >
> > > Related work and novelty
> > >
> >
> > On related work and novelty, my concern was not that Accordion-Thinking is identical to prior work, but that **the paper does not clearly articulate the distinction**. In Sec. 2.2, after discussing LighterThinker and Delethink, the paper states: “Unlike previous approaches, ... allows the model to perform progressive step-by-step reasoning by retaining the summary of its current reasoning at each step before proceeding further.” This high-level description also applies to methods such as LightThinker and InftyThink. The rebuttal clarifies the intended differences, but those differences need to be stated clearly in the paper itself.
> >
> > Relatedly, Sec. 3.2 describes a teacher-rewrite pipeline that converts long CoT traces into segmented reasoning with step summaries, followed by rule-based filtering. **This pipeline is very similar to LighterThinker/InftyThink, and these works are not credited in the paper for the data pipeline.**
> >
> > > Empirical results
> > >
> >
> > On the empirical side, my review did not ask for Fold to outperform Unfold. The question was **whether the paper has established that standard SFT or standard RL truly degrades Fold performance in a reliable way, or whether the observed gaps could plausibly be evaluation noise**. In the main paper, the claims that “SFT is insufficient” and that standard RL still “neglects compression” are supported by relatively small gaps in Table 1 (especially when using small datasets like AIME, AMC where 2-3% amounts to one question). These could very well be variance due to stochasticity, and since the results are provided on a single seed, I do not think the strength of this claim is established. The added OOD numbers in the rebuttal also show small differences between unfold and fold methods, further strengthening this concern.
> >
> > > Delethink
> > >
> >
> > I appreciate the added external baselines in the rebuttal. At the same time, this reinforces my original concern that the submission itself omitted important comparisons. Since both this work and Delethink use RL to train policies on compact contexts, **I still view Delethink as one of the most relevant missing baselines** for evaluating whether readable summaries matter for performance, rather than only for interpretability.
> >
> > Finally, I do not find the attention-map analysis very informative in its current form. The higher value of the attention weights are trivially a result of shorter contexts (due to the softmax normalization), so I am not really sure what is being claimed or demonstrated here. The same analysis on any shorter context (whether truncated states of Delethink or even gibberish text) would show higher attention weights simply because they are shorter length.
> >
> > The central claim of the paper is that standard SFT or RL hurts Fold-mode performance, and that the proposed method removes this gap. **I do not think the current evidence meets the standard needed to support that claim. The reported differences are small and based on a single seed, which means that the observed gaps could plausibly be within evaluation noise. Because this issue is central to the paper’s motivation, I maintain my score.**

---

> > > ### Author Response · Authors · 2026-04-03
> > >
> > > We appreciate the follow-up, but respectively disagree that our central point can be explained as evaluation noise.
> > >
> > > ## **1. Related work / novelty.**
> > > We do not claim that the generic summarize-and-continue paradigm as our core contribution. LightThinker/InftyThink are relevant summarize-and-continue baselines, and Delethink is a compact-state RL baseline. Our claim is narrower: Accordion-Thinking uses **model-generated readable step summaries as the carried reasoning state**, and this state is learned mainly through **Fold-RL / Mix-RL**, rather than being fixed by an SFT template or a deletion rule. In the revision, we will expand Secs. 2.2/3.2 to make this methodological gap explicit while also crediting prior work in the cold-start section. (Due to ICMLrebuttal space limits, and the anonymous links cannot expose revised textual sections, we cannot show the edited wording directly here)
> > >
> > > ## **2. Empirical significance.**
> > > Table 1 uses **n=32 sampling**. We report rollout-level pass@1 averaged over 32 samples per problem, not question-level pass@k where 2-3% means one question. Thus the deltas correspond to many failed rollouts, not a single noisy item. This protocol is standard in recent math reasoning [1,2,3]. The gaps also persist on MATH500 and Minerva. Importantly, the OOD results in our previous rebuttal were **after RL**, so the small gap there is expected: RL is meant to reduce it. Before RL, the cold-start OOD gap is much larger:
> > >
> > > 7B: ARC-C Unfold/Fold = 77.2/73.2, GPQA-D Unfold/Fold = 37.9/32.1
> > >
> > > 4B: ARC-C Unfold/Fold = 86.3/83.1, GPQA-D Unfold/Fold = 43.0/37.5
> > >
> > > (We will add this to the OOD table.)
> > >
> > > To address seed variance directly, we added extra seed sweeps; each SFT-seed is evaluated with 3 random seed, results are listed below. In our runs, the variance is very small (most of the gap is less than **0.5%**), so the Fold/Unfold behavior is stable rather than a seed artifact.
> > >
> > > |Model|Variant|sft seed|AIME24|AIME25|MATH500|Minerva|AMC|
> > > |-|-|-|-|-|-|-|-|
> > > |Q2.5-7B|Unfold|42|26.7-26.9|24.4-24.8|86.2-86.5|65.3-65.8|39.6-39.9|
> > > |||24|26.5-26.9|24.6-24.6|86.4-86.8|65.5-65.5|39.4-39.9|
> > > ||Fold|42|22.8-23.0|22.8-23.5|82.1-82.4|62.2-62.5|37.2-37.6|
> > > |||24|23.1-23.3|22.4-22.8|82.0-82.3|62.1-62.4|37.4-37.9|
> > > |Q3-4B|Unfold|42|23.6-24.1|25.4-25.6|84.7-85.0|63.9-64.4|39.3-40.0|
> > > |||24|23.8-24.0|25.6-26.0|84.8-85.2|63.8-64.4|39.6-39.9|
> > > ||Fold|42|18.6-19.2|21.9-22.2|79.0-79.3|57.1-57.4|35.3-35.6|
> > > |||24|18.9-19.3|21.8-22.4|79.0-79.2|57.0-57.3|35.1-35.4|
> > >
> > > ```
> > > [1]DeepSeek-R1... Nature 2025
> > > [2]SimpleRL-Zoo... COLM 2025
> > > [3]Laser... ICLR 2026
> > > ```
> > >
> > > ## **3. Delethink.**
> > > This baseline was already running during our previous response, and we apologize that we could not report it in time; the runs finished too late to organize before the rebuttal deadline. We now provide it under the **same cold-start initialization** and **matched step-response length**:
> > >
> > > |Model|Method|A24|A25|M500|AMC|Mine|Macro|Peak|
> > > |-|-|-|-|-|-|-|-|-|
> > > |Q2.5-7B|Delethink|31.0|26.9|89.3|**72.5**|**42.0**|52.3|5022|
> > > ||Accordion Mix-RL|**32.2**|**28.3**|**89.6**|71.9|41.8|**52.8**|**3533**|
> > > |Q3-4B|Delethink|25.5|26.7|**89.2**|72.7|**43.6**|51.5|6329|
> > > ||Accordion Mix-RL|**27.6**|**28.0**|88.6|**72.8**|43.4|**52.1**|**3925**|
> > >
> > > Delethink is competitive on MATH500/Minerva/AMC, but weaker on AIME24/25, and its PeakToken is higher because it keeps half of the response chunk size rather than compressing it into a short explicit state.
> > >
> > > Furthermore, the cold-start comparison against Unfold is bellow. (Delethink also starts with a clear gap to Unfold)
> > > |Model|Mode|A24|A25|M500|AMC|Mine|Macro|
> > > |-|-|-|-|-|-|-|-|
> > > |Q2.5-7B|Unfold|26.7|24.6|86.2|65.4|39.7|48.5|
> > > ||Delethink|22.5|21.1|83.7|60.6|38.4|45.3|
> > > |Q3-4B|Unfold|23.8|25.4|84.7|64.1|39.5|47.5|
> > > ||Delethink|18.8|15.4|80.9|55.0|37.9|41.6|
> > >
> > > ## **4. Attention map**
> > > Our plot is not a raw max-attention visualization. Given input prefix $x_{1:m}$ and continuation $y_{1:n}$, we run one forward pass on $[x;y]$, average causal attention over heads and then over layers to obtain $\bar{A}$, and assign each input token $x_i$ the heat score $H_i=\frac{1}{n}\sum_{t=1}^{n}\bar{A}_{m+t,i}$. Thus the blue intensity is the **average attention mass sent by later generated tokens back to earlier context tokens**. In Accordion, these later-token-derived scores become denser on the compact carried summaries; we interpret this as evidence of information compression, not as a standalone proof.
> > >
> > > ---
> > > ## **Conclusion**
> > > - **`Rebuttal experiments will be integrated into the revision.`** The revised main table is in [[Rebuttal-Fig5]](https://anonymous.4open.science/r/Accordion-ICML-Rebuttal-3D5D/Rebuttal-Fig5.pdf).
> > > - **`The Fold/Unfold gap is not seed-driven evaluation noise`**, and we observe the same phenomenon for Delethink-style generation.
> > > - **`Both RL strategies can reduce this gap, but Accordion-Thinking is more efficient.`** Compared with Delethink RL, it achieves substantially lower PeakToken while also providing readable step summaries.

---

### Official Review · Reviewer_MzkL · 2026-03-13

**Soundness:** 3
**Presentation:** 3
**Significance:** 2
**Originality:** 2
**Overall Recommendation:** 4
**Confidence:** 3

**Summary:**

This paper introduces an LLM reasoning framework called Accordion-Thinking, which is designed to resolve the computational complexity and memory bottlenecks associated with long CoT reasoning. It achieves this by teaching the model to periodically summarize and discard redundant historical steps during the reasoning process. Consequently, this method maintains solution accuracy and achieves a significant increase in throughput.

**Compliance With Llm Reviewing Policy:**

Affirmed.

**Final Justification:**

I thank the authors for their detailed reply. I will maintain my score and positive recommendation!

**Key Questions For Authors:**

1. Clarification on Related Ideas: Both MEM1[1] and Agent-Fold[2] focus on context management. Aside from the differences in evaluation tasks, what is the fundamental methodological difference between your approach and these existing methods?
2. Missing SFT-RL baseline: The paper compares its method against Zero-RL, which applies RL directly to the base model. However, a critical missing comparison is an "SFT + RL" baseline, where the base model is first fine-tuned using the unoverwritten SFT data before undergoing RL. How would the proposed approach perform when compared against this baseline?
3. Reporting Average Reasoning Steps: The proposes the multi-step reasoning process. What is the average number of reasoning steps (or "folds")?
4. Token Consumption Metrics: While the paper successfully demonstrates improved throughput, could the authors provide a detailed result of the overall input and output token consumption?

[1]MEM1: https://openreview.net/forum?id=XY8AaxDSLb
[2]Agent-Fold: https://openreview.net/forum?id=IuZoTgsUws

**Limitations:**

yes

**Strengths And Weaknesses:**

## Strengths
- Clear motivation: It aims to address the high computational complexity and heavy memory footprint caused by long Chain-of-Thought reasoning.
- Simple but effective Method: This paper breaks down the long CoT process into discrete steps and manages the context for each step, achieving higher throughput.

## Weakness
- Rely on High-Qualiy SFT data: The paper utilizes DeepSeek-V3.2 as the teacher model to generate SFT data. Employing a weaker model may lead to lossy information compression and compromised model performance.
- Lack of Baseline Comparisons: Although the authors discuss related methods such as LighterThinker and Delethink in the Related Work section, the experimental evaluation lacks a direct empirical comparison of performance and efficiency against these established baselines.

---

> ### Author Rebuttal · Authors · 2026-03-30
>
> Thank you for your constructive feedback and for acknowledging the clear motivation and simple yet effective design of our Accordion-Thinking framework. We appreciate your valuable suggestions, which have helped us further strengthen our paper. Below, we address your concerns and questions in detail.
>
> ---
> ## **Response to Weaknesses**
> ### **W1: Reliance on High-Quality SFT data.**
> We agree that the quality of SFT data plays a role in the initial "cold-start" phase, primarily for teaching the model the structural format of our step-summarization. However, the contribution of our framework includes **Reinforcement Learning (RL) fundamentally bridges the gap of lossy information compression**. As demonstrated in our "Gap-Vanishing" phenomenon (Fig 3), the RL process teaches the model to self-regulate and perfectly encode essential states into summaries, recovering the performance drop observed in the SFT stage.
>
> To further address your concern, we conduct an experiment using our own model (Qwen2.5-Math-7B with Mix RL Training) to generate the initial SFT data. Benefiting from the model's self-regulated step-by-step CoT, its SFT results even outperform those derived from synthetic data. The results are shown in the following:
>
> |Model|SFTDataSynthesizer|AIME24|MATH500|AMC|
> |-|-|-|-|-|
> |Qwen2.5-Math-7B|DS-V3.2-Syn-14k|23.0|82.3|62.4|
> ||MixRL-Gen-7k|**29.5**|**86.3**|**66.6**|
> |Qwen3-4B-Base|DS-V3.2-Syn-14k|19.2|79.2|57.3|
> ||MixRL-Gen-7k|**22.5**|**84.7**|**59.7**|
>
>
> ### **W2: Lack of Baseline Comparisons.**
> Following your suggestion, we have expanded our empirical evaluation to include direct comparisons with established efficient reasoning baselines, specifically LighterThinker and H2O.
>
> |Model|Method|AIME24|AIME25|MATH500|AMC|Minerva|Macro|
> |-|-|-|-|-|-|-|-|
> |Qwen2.5-Math-7B|Unfold-RL(SFT+RL)|32.0|26.7|89.2|71.2|**42.1**|52.2|
> ||H2O|25.1|20.3|82.9|61.5|34.7|44.9|
> ||LightThinker|27.2|22.5|84.4|67.7|37.0|47.8|
> ||Mix-RL(ours)|**32.2**|**28.3**|**89.6**|**71.9**|41.8|**52.8**|
> ||
> |Qwen3-4B-Base|Unfold-RL(SFT+RL)|27.5|27.8|**88.9**|**73.2**|42.5|52.0|
> ||H2O|22.7|20.9|80.4|64.7|34.6|44.7|
> ||LightThinker|23.9|23.2|84.0|67.7|39.0|47.6|
> ||Mix-RL(ours)|**27.6**|**28.0**|88.6|72.8|**43.4**|**52.1**|
>
> Although H2O and LightThinker can accelerate inference, their non-model-autonomous folding mechanism impairs the model's inference performance, which our approach avoids.
>
> ---
> ## **Response to Key Questions**
> ### **Q1: Differences from MEM1 and AgentFold**
> We thank the reviewer for highlighting these works, and will cite the papers in Related Works. While all share the concept of "context management," they differ fundamentally in scope and methodology:
> 1. **Task Setting:**
>    MEM1 and AgentFold are designed for multi-turn interactive environments (e.g., web agents), managing external observations and tool calls. In contrast, Accordion-Thinking targets single-turn, long-form internal CoT reasoning, specifically addressing the KV-cache and attention bottlenecks in continuous, self-generated monologues.
> 2. **Methodology (Rule-driven vs. Self-regulated RL):**
>    * MEM1 updates its memory discretely upon receiving external turn-based feedback. Our model autonomously determines *when* to segment and summarize its continuous thoughts without relying on external turn prompts.
>    * AgentFold relies on SFT to emit rigid, rule-based folding directives (e.g., JSON step ranges). Our approach employs an end-to-end RL framework that naturally internalizes lossless compression, achieving dynamic self-regulation without complex explicit state management.
> ### **Q2: Missing SFT-RL baseline.**
> We sincerely apologize for the naming ambiguity in our paper, which caused this misunderstanding. **The "Unfold-RL" baseline in our paper IS exactly the "SFT + RL" baseline you mentioned.**
> Specifically, both `Unfold-RL`, `Fold-RL`, and `Mix-RL` models are initialized from the exact same unoverwritten SFT model. `Unfold-RL` applies RL directly on this SFT model without context compression.
> ### **Q3 & Q4: Average Reasoning Steps and Token Consumption.**
> To provide a clearer picture of our efficiency, we break down the average folding steps and token consumption (avg on all benchmarks):
>
> |Model|Method|PeakToken|AvgFolds(Steps)|AvgGenLength|
> |-|-|-|-|-|
> |**Qwen2.5-Math-7B**|UnFold-RL(SFT+RL)|6896|-|6544|
> ||Mix-RL(UnFoldMode)|6523|-|6244|
> ||**Mix-RL(FoldMode)**|**2853**|3.57|6712|
> |**Qwen3-4B-Base**|UnFold-RL(SFT+RL)|8512|-|8231|
> ||Mix-RL(UnFoldMode)|8311|-|8096|
> ||**Mix-RL(FoldMode)**|**2974**|4.07|8532|
>
> As shown above, while the total gen length slightly increase due to the explicit generation of step summaries, **the active context window (`Peak Token`), which dictates the actual GPU memory and KV-cache bottleneck, is drastically reduced.**
> Furthermore, we show the training dynamics of the average steps and token consumption in [[Rebuttal-Fig1]](https://anonymous.4open.science/r/Accordion-ICML-Rebuttal-3D5D/Rebuttal-Fig1.pdf).

---

> > ### Author Rebuttal · Reviewer_MzkL · 2026-04-03
> >
> > I thank the authors for their detailed reply. I will maintain my score and positive recommendation!

---

> > > ### Author Response · Authors · 2026-04-03
> > >
> > > Dear Reviewer MzkL,
> > >
> > > Thank you very much for your acknowledgment of our rebuttal.
> > >
> > > We noticed that you updated the response status to "(c) Partially resolved or unresolved...", indicating that you still have some concerns. We take this feedback very seriously, as our goal is to ensure the utmost technical rigor and clarity of this work.
> > >
> > > However, since your latest comment does not specify your concerns, we are currently at a loss as to how to best address your feedback.
> > >
> > > We would sincerely appreciate it if you could provide even a clarification on your remained concerns. We are fully committed to providing further evidence, additional experiments, or detailed explanations during this discussion period to resolve any lingering doubts you may have. If you feel that our response has resolved all your concerns, we would like to kindly ask you to consider revising your overall recommendation or confidence score.
> > >
> > > We look forward to hearing from you and are ready to engage in further discussion.
> > >
> > > Best regards,
> > >
> > > Authors of Paper 20603

---

### Decision · Program_Chairs · 2026-04-30

**Decision:**

Accept (regular)

**Comment:**

Accordion-Thinking introduces a framework for efficient reasoning by teaching LLMs to self-regulate step granularity through dynamic summarization. Reviewers Lid5 and MzkL strongly endorsed the practical 3x throughput gains and the reduction in KV cache overhead, which significantly improves real-world deployment potential. Reviewer ZeP1 highlighted the scientific importance of the gap-vanishing phenomenon, where RL training allows compressed reasoning to eventually match the accuracy of full-context chains. Although Reviewer Q1Kd initially raised concerns about evaluation noise and novelty, the authors' detailed rebuttal—including n=32 sampling, seed variance analysis, and OOD results on GPQA-Diamond—successfully established the method's robustness and its edge over baselines like Delethink and H2O. The transition from rule-based context management to a model-native, learnable capability was recognized as a major contribution. Given that all the reviewers are positive about this paper, I would recommend acceptance.